# Rainfall retrieval algorithm for commercial microwave links: stochastic calibration

Wagner Wolff[1], Aart Overeem[2,3], Hidde Leijnse[2,3], and Remko Uijlenhoet[3,4]

[1]Department of Biosystems Engineering, University of São Paulo/"Luiz de Queiroz" College of Agriculture (ESALQ/USP);
[2]R&D Observations and Data Technology, Royal Netherlands Meteorological Institute (KNMI);
[3]Hydrology and Quantitative Water Management Group, Wageningen University & Research (WUR);
[4]Department of Water Management, Delft University of Technology (TU Delft).

**Correspondence:** Wagner Wolff (wwolff@usp.br)

**Abstract.** During the last decade, rainfall monitoring using signal level data from commercial microwave links (CMLs) in cellular communication networks has been proposed as a complementary way to estimate rainfall for large areas. Path-averaged rainfall is retrieved between the transmitting and receiving cellular antenna of a CML. One rainfall estimation algorithm for CMLs is RAINLINK, which has been employed in different regions (e.g., Brazil, Italy, the Netherlands, and Pakistan) with satisfactory results. However, the RAINLINK parameters have been calibrated for a unique optimum solution, which is inconsistent with the fact that multiple similar or equivalent solutions may exist due to uncertainties in algorithm structure, input data, and parameters. Here, we show how CML rainfall estimates can be improved by calibrating all parameters of the algorithm systematically and simultaneously with the stochastic optimization method Particle Swarm Optimization, which is used for the numerical maximization of the objective function. An open dataset of approximately 2,800 sub-links of minimum and maximum received signal levels over 15-minute intervals covering the Netherlands ($\sim$35,500 km$^2$) is employed, where 12 days are used for calibration and 3 months for validation. A gauge-adjusted radar rainfall dataset is utilized as reference. Verification of path-average daily rainfall shows a reasonable improvement for the stochastically calibrated parameters with respect to RAINLINK's default parameter settings. Results further improve when averaged over the Netherlands. Moreover, the method provides a better underpinning of the chosen parameter values and is therefore of general interest for calibration of RAINLINK's parameters for other climates and cellular communication networks.

## 1  Introduction

Accurate rainfall observations with high temporal and spatial resolution are crucial for, e.g., agriculture, meteorology, flood warnings and fresh water resource management. However, for many places on the earth's land surface, accurate rainfall information is lacking, especially from ground-based measurements at sub-daily and daily time scales (Sun et al., 2018). Another issue is the data availability of ground-based measurements. For instance, the largest worldwide rain gauge database, maintained by the Global Precipitation Climatology Centre (GPCC) had 45,000 rain gauges in 1961-2000 and down to 10,000 after 2016. This decrease was caused by a delay in data delivery and by post-processing at GPCC (Schneider et al., 2021). Although,

decreasing in the past due to quality control, the GPCC database has been increasing in recent years as a result of delivery of updates as well as supplements with additional stations and long time-series of data (Schneider et al., 2021).

Suggested by Upton et al. (2005) and initially applied by Messer (2006) and Leijnse et al. (2007), the technique to esti-mate rainfall intensities based on signal level data from commercial microwave links (CMLs) is slowly but surely becoming a complementary source of rainfall information next to traditional ground-based measurements from rain gauges, weather radars and disdrometers. A CML is the link along a path between a transmitting antenna on one cell phone tower and a receiving antenna on another cell phone tower, often having two sub-links for communication in both directions. Since rainfall attenuates

microwave radiation at frequencies of tens of GHz (wavelengths of about 1 cm), typically employed by CMLs, the integrated rain-induced attenuation along the link path can be computed from the decrease in signal levels with respect to dry weather, and subsequently converted to path-average rainfall. The core of the rainfall retrieval algorithm is the conversion of specific attenuation $k$ (dB km$^{-1}$) to path-average rainfall intensity $R$ (mm h$^{-1}$) via the power-law relation $R = ak^b$ (Atlas and Ulbrich, 1977; Olsen et al., 1978). The coefficient $a$ (mm h$^{-1}$ dB$^{-b}$ km$^b$) and exponent $b$ (–) depend mainly on the microwave link's

frequency and polarization and on the rain drop size distribution (DSD) (Leijnse et al., 2007). Before applying the power-law relation, the received signal power must be processed to filter out any attenuation unrelated to rain, and to compare signals dur-ing a rainy interval with those from dry intervals. A typical workflow consists of: (i) CML data acquisition and preprocessing; (ii) identification of rain events in noisy raw data (wet-dry classification); (iii) baseline determination, representative of dry intervals; (iv) removal of outliers due to malfunctioning links; (v) correction of received signal powers; and (vi) computation

of mean path-average rainfall intensities (Overeem et al., 2016a; Chwala and Kunstmann, 2019).

    An advantage of CMLs is that they use the existing infrastructure of mobile network operators (MNOs) for network mainte-nance, data storage and acquisition. Furthermore, CMLs can be employed as a complement to existing rain gauge and weather radar networks, as well as in areas where instruments for ground observation are poor or non-existent. Thus, rainfall retrieval from CML data and subsequent mapping is a form of "opportunistic" sensing that has gained prominence in recent years

(Uijlenhoet et al., 2018; Chwala and Kunstmann, 2019).

    A number of studies highlight the successful employment of CMLs for rainfall retrieval, of which the most relevant for this study are discussed here. Zinevich et al. (2009) show that this technique is suitable for measuring near-ground rainfall around the cities of Ramle and Modi'in (area $\approx$ 900 km$^2$; density $\approx$ 0.025 CML km$^{-2}$) in Israel. Incorporating the uncer-tainty associated with the different sources of rainfall information, Bianchi et al. (2013) obtained reliable rainfall intensity

estimates by combining rain gauge, radar, and microwave link observations in the Zürich area, Switzerland (area $\approx$ 460 km$^2$; density $\approx$ 0.03 CML km$^{-2}$). In a dedicated case study in Prague, Czech Republic, Fencl et al. (2015) used 14 CMLs over a small area of 2.5 km$^2$ (i.e. a density of 5.6 CML km$^{-2}$), concluding that quantitative precipitation estimates from CMLs capture the spatio-temporal rainfall distribution at the microscale very well. Recently, de Vos et al. (2019) reached correlations around 0.60 for daily rainfall accumulations, using instantaneously sampled data from a CML network in the Netherlands

(density $\approx$ 0.054 CML km$^{-2}$). Moreover, comparing those results with earlier studies in the Netherlands, the authors highlight min/max sampling outperforms instantaneous sampling in terms of rainfall estimates. Long-term studies involving country-wide verification of CML rainfall estimates based on data from a few thousand CMLs are provided by Overeem et al. (2016b)

for the Netherlands employing RAINLINK (Overeem et al., 2016a), and by Graf et al. (2020) for Germany employing py-comlink (https://github.com/pycomlink/pycomlink), both open-source rainfall retrieval packages. Machine learning supervised algorithms have been used for rainfall retrieval via CMLs, improving the performance of this kind of rainfall measurement (Pudashine et al., 2020; Habi and Messer, 2021). These data-driven solutions also hold a promise for ungauged areas, but it will not be feasible for places or countries without sufficient reference data to train the machine learning algorithms. That is, data-driven models require a huge number of observations to learn and detect the whole behavior of the phenomenon to be modeled. For other algorithms, such as RAINLINK, it may still be feasible to at least tune a few parameters, for instance, by employing drop size distribution observations (from a region with a similar climate) to obtain more appropriate coefficients of the relationship between specific attenuation and rain rate.

Likewise, research has been conducted to evaluate CML-derived rainfall in hydrological model responses. Brauer et al. (2016) study the effects of differences in rainfall measurement techniques (including CMLs) on discharge and groundwater simulations using a lumped rainfall-runoff model in a small (6.5 km$^2$) catchment. CML-derived rainfall estimates are found to lead to much better results than real-time weather radar data when comparing discharge and groundwater simulations to observations for a full year. Investigating the potential of CML-derived rainfall estimates for streamflow prediction and water balance analyses, Smiatek et al. (2017) observe a significant improvement in the reproduction of observed discharge values for events with local heavy rainfall. The authors find that even rainfall fields provided by gauge-adjusted weather radar do not capture such events, which suggests that an extremely dense monitoring network would be needed to properly capture local heavy rainfall. Likely, this explains why Liberman et al. (2014) achieve better results by merging CML and radar data rather than using just one of these sources to retrieve rainfall intensities.

Despite all these studies showing the potential of CMLs for rainfall monitoring, challenges remain. These are mainly related to dealing with typical sources of error, e.g., wetting of antennas in rain events causing additional attenuation and hence resulting in rainfall overestimation, as well as signal level decrease during dry periods in CML raw data (Leijnse et al., 2008; Messer and Sendik, 2015; Overeem et al., 2016a). Rainfall retrieval algorithms for CMLs aim to take these phenomena into account, although issues such as wet-dry classification still require improvement. Another challenge concerns the calibration of the parameters of the rainfall retrieval algorithms. Current calibration procedures fall short of addressing the uncertainties associated with CML signal levels (e.g. due to different brands of antennas and varying path lengths), algorithm structure (e.g. attenuation thresholds for classification of rainy and non-rainy periods), model parameters (e.g. for wet antennas and outlier filters), and rainfall itself (e.g. due to DSD spatial variability along the link path). Concretely, the parameters of the algorithms are calibrated empirically in order to obtain a unique optimum solution. In fact, many optimum solutions can occur, corresponding to different parameters sets (a phenomenon known as equifinality).

Here, we partly address this by calibrating the most important parameters of the open-source rainfall retrieval algorithm RAINLINK systematically and simultaneously with the stochastic optimization method Particle Swarm Optimization. This is preceded by a sensitivity analysis selecting the most important parameters. RAINLINK has been used for CML rainfall estimation in various regions, i.e. Australia, Brazil, Italy, the Netherlands, Nigeria, Pakistan, and Sri Lanka (Overeem et al., 2016a, b; Sohail Afzal et al., 2018; Rios Gaona et al., 2018; de Vos et al., 2019; GSMA, 2019; Roversi et al., 2020; Overeem

et al., 2021b; Pudashine et al., 2021), and has been calibrated deterministically (Overeem et al., 2011, 2013, 2016a, b; de Vos et al., 2019). With the new optimization method, we provide a better underpinning of parameter values for this CML rainfall retrieval algorithm. Moreover, we optimize for the first time the main RAINLINK processes, i.e., wet-dry classification and rainfall retrieval, separately. These resulting CML rainfall estimates are contrasted to those based on RAINLINK's default parameter values (Overeem et al., 2011, 2013, 2016a). A gauge-adjusted radar rainfall dataset is utilized as reference for the CML-derived path-average rainfall estimates. We use a large publicly available CML dataset of approximately 2,800 sub-links of minimum and maximum received signal levels over 15-minute intervals covering the Netherlands ($\sim$ 35,500 km$^2$), where 12 days are used for calibration and 3 months for validation.

This study is organized as follows. First, study area and datasets (Section 2.1), and methodology (Sections 2.2 and 2.3) employed for RAINLINK calibration are presented. Next, the results and discussion (Section 3) present our major findings. Finally, the conclusions (Section 4) summarize the findings and highlight the recommendations and outlooks for further research.

## 2 Material and methods

### 2.1 Study area and datasets

The study area considered is the Netherlands ($\sim$ 35,500 km$^2$; Fig. 1 (a)), which has a temperate oceanic climate according to the Köppen-Geiger classification (Peel et al., 2007). CML data were obtained from MNO T-Mobile NL: minimum and maximum received powers over 15-minute intervals, based on 10 Hz sampling with a precision of 1 dB. Data from approximately 2,800 sub-links (validation) and 2,940 (calibration) per time interval were available (after preprocessing with RAINLINK). The 12-day calibration dataset, used to optimize RAINLINK's parameters, covers the period from June to September 2011. It served as validation dataset in Overeem et al. (2013). The 3-month validation dataset covers the summer months June, July, and August 2012. We are only using data from summer in the Netherlands to prevent analyzing events with solid precipitation. This has the added advantage of the data bearing greater resemblance to rainfall in (sub)tropical climates, where the use of CMLs for rainfall retrieval has the largest potential.

Figure 1 illustrates the main characteristics of the CML dataset used for validation. Being distributed over the entire country (Fig. 1 (a)), the CMLs have a high temporal and spatial data availability, i.e., 92% of sub-links have observations for more than 80% of the period. In spite of not having a perfectly uniform distribution in terms of their directions, all direction classes are well-represented (Fig. 1 (b)). Microwave frequencies range from $\sim$13 GHz to 40 GHz (the majority from 37 to 40 GHz, Fig. 1 (c)). Lengths vary from 0.1 km to 20 km (the majority less than 5 km, Fig. 1 (d)), where shorter lengths typically correspond to higher microwave frequencies (Fig. 1 (d)).

A climatological gauge-adjusted radar rainfall dataset of 5-min rainfall depths, aggregated over 15 minutes, was used as reference for calibration of the rainfall retrieval algorithm (RAINLINK) and validation of rainfall estimates. The radar dataset is maintained by the Royal Netherlands Meteorological Institute (KNMI) and has a 1-km spatial resolution. For more details see Overeem et al. (2009a, b, 2011).

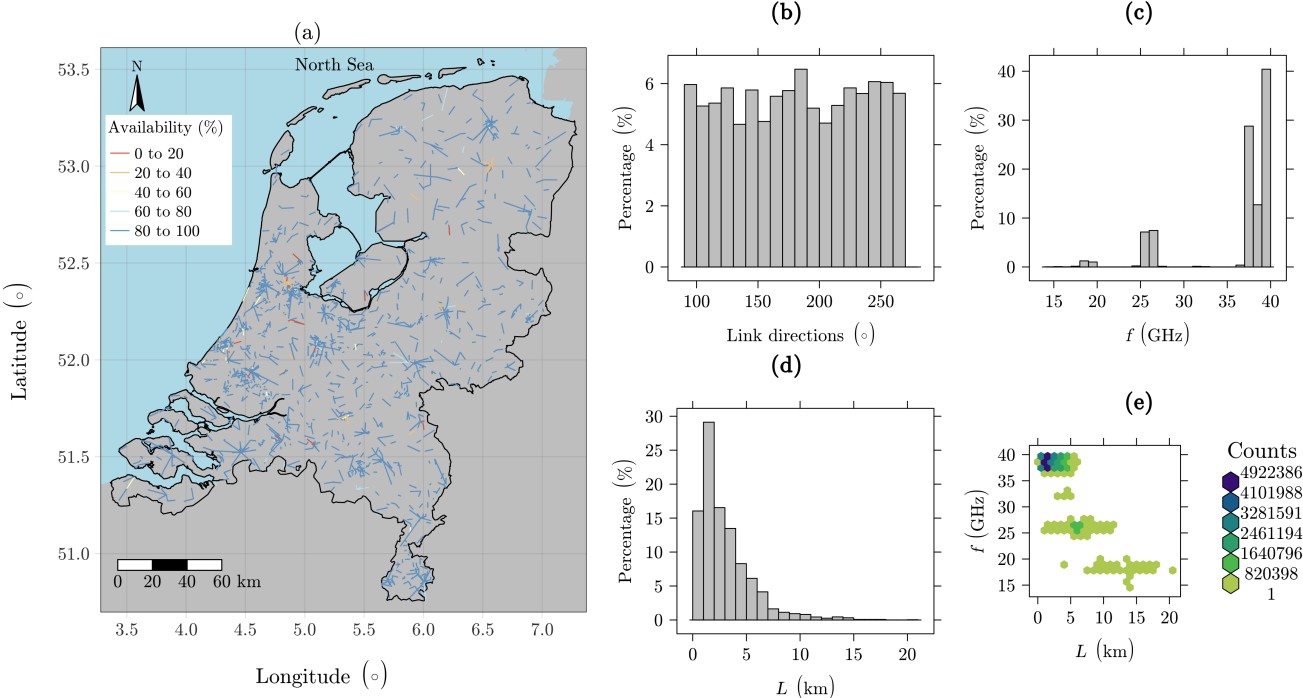

**Figure 1.** Map of the Netherlands and sub-link characteristics for the validation dataset: (a) CML locations and availability, (b) distribution of link directions, (c) distribution of microwave link frequencies, (d) distribution of link lengths, and (e) density of link length and frequency combinations.

## 2.2 Rainfall retrieval algorithm

Overeem et al. (2016a) describes the CML rainfall retrieval algorithm RAINLINK. Made available as R-package (R Core Team, 2018), current version of RAINLINK is 1.21 (Overeem et al., 2021a), and version 1.2 was used in this study, which is hosted at GitHub. RAINLINK's default parameter values are derived or selected in (Overeem et al., 2011, 2013, 2016a). The algorithm begins with a quality control by preprocessing of the CML data. Links with frequencies lower than 12.5 GHz and higher than 40.5 GHz are discarded. Moreover, the attributes frequency, link coordinates, path length, and identifier are checked for either duplicated or mismatches of information. Next, RAINLINK is divided into two main sub-processes, one for defining wet and dry periods and the other one for the actual rainfall retrieval.

### 2.2.1 Wet-dry classification

The process to define wet and dry periods assumes that rainfall is spatially correlated. Therefore, during a rainy time interval, a substantial decrease in received signal levels should be detected by nearby links within a specific radius (Overeem et al., 2011, 2016a). This approach is called "nearby link approach". The output is a binary response to indicate wet and dry periods, respectively. Table 1 highlights all employed parameters.

**Table 1.** Wet-dry classification parameters (WD$_{pn}$): default values from RAINLINK and calibration search space (minimum and maximum values). Modified from Overeem et al. (2016a).

| Parameter description | Symbol and unit | Default value | Minimum value | Maximum value |
|---|---|---|---|---|
| WD$_{p1}$ – Minimum number of hours needed to compute max($P_{min}$) | – (h) | 6 | 2 | 15 |
| WD$_{p2}$ – Number of previous hours over which max($P_{min}$) is to be computed (also determines period over which cumulative difference $F$ of outlier filter is computed) | – (h) | 24 | 6 | 24 |
| WD$_{p3}$ – Radius | $r$ (km) | 15 | 10 | 30 |
| WD$_{p4}$ – Attenuation threshold | median($\Delta P$) (dB) | $-1.4$ | $-8$ | 0 |
| WD$_{p5}$ – Specific attenuation threshold | median($\Delta P_{L}$) (dB km$^{-1}$) | $-0.7$ | $-2$ | 0 |
| WD$_{p6}$ – Minimum number of available (surrounding) links | – (–) | 3 | 3 | 10 |
| WD$_{p7}$ – Minimum received power threshold | – (dB) | 2 | 1 | 4 |

### 2.2.2 Rainfall retrieval

Once the rainy and non-rainy time intervals have been identified, a reference signal level ($P_{ref}$) is computed, which represents the median received power during dry intervals. Next, outliers are removed by applying a filter which uses specific attenuation derived from the uncorrected minimum received power. It assumes that rainfall is correlated in space. The filter removes a time interval of a link for which the time-integrated difference between its specific attenuation and that of the surrounding links over the previous period (Table 2, parameter RR$_{p2}$) is lower (i.e. more negative) than a certain threshold (Table 2, parameter RR$_{p3}$).

Preventing non-zero rainfall estimates during non-rainy intervals, corrected minimum ($P_{min}^{C}$) and maximum ($P_{max}^{C}$) received powers are calculated by adjusting the signals to the base level for non-rainy intervals. Subsequently, the minimum and maximum rain-induced attenuation, $A_{min}$ (dB) and $A_{max}$ (dB), respectively, are calculated for each link and time interval using

$$A_{min} = P_{ref} - P_{max}^{C},$$
$$A_{max} = P_{ref} - P_{min}^{C}. \tag{1}$$

Next, the minimum and maximum path-averaged rainfall intensities, $\overline{R}_{min}$ (mm h$^{-1}$) and $\overline{R}_{max}$ (mm h$^{-1}$), respectively, are

computed according to

$$\overline{R}_{min,\,max} = a \left( \frac{A_{min,\,max} - A_{a}}{L} H \left( A_{min,\,max} - A_{a} \right) \right)^{b}, \tag{2}$$

where $H$ is the Heaviside function (if the argument of $H$ is smaller than zero, $H = 0$; else $H = 1$). $A_{a}$ (dB) is a fixed wet antenna attenuation correction term, and $a$ (mm h$^{-1}$ dB$^{-b}$ km$^{b}$) and $b$ (-) are the coefficient and exponent of the employed power-law $R - k$ relation, respectively. The values of $a$ and $b$, which depend mainly on link frequency, have been derived from

measured raindrop size distributions and computations of electromagnetic scattering by rain drops for vertically polarized

**Table 2.** Rainfall retrieval parameters (RR$_{pn}$): default values from RAINLINK and calibration search space (minimum and maximum values). Modified from Overeem et al. (2016a).

| Parameter description | Symbol and unit | Default value | Minimum value | Maximum value |
|---|---|---|---|---|
| RR$_{p1}$ – Minimum number of hours that should be dry in preceding period | – (h) | 2.5 | 2.5 | 12 |
| RR$_{p2}$ – Period over which reference level is to be determined | – (h) | 24 | 12 | 24 |
| RR$_{p3}$ – Outlier filter threshold | $F_t$ (dB km$^{-1}$ h) | −32.5 | −100 | 0 |
| RR$_{p4}$ – Wet antenna attenuation | $A_a$ (dB) | 2.3 | 0 | 5 |
| RR$_{p5}$ – Temporal rainrate distribution coefficient | $\alpha$ (–) | 0.33 | 0.1 | 0.6 |

signals (Leijnse et al., 2008). The polarization for individual links was unknown, but the majority of links used vertically polarized signals.

Finally, the mean path-averaged rainfall intensity, $\overline{R}$ (mm h$^{-1}$) is computed by means of

$$\overline{R} = \alpha\overline{R}_{\mathrm{max}} + (1-\alpha)\,\overline{R}_{\mathrm{min}}, \tag{3}$$

where $\alpha$ is a coefficient which determines the contribution of the minimum and maximum path-averaged rainfall intensity during a time interval. Table 2 gives an overview of all parameters used in the rainfall retrieval process.

### 2.3 RAINLINK sensitivity analysis and calibration

Using the mean 15-min path-averaged rainfall intensities retrieved from RAINLINK, the parameters with the highest importance in the algorithm are identified by means of a sensitivity analysis called Latin-Hypercube One-factor-At-a-Time (LH-OAT)
(Van Griensven et al., 2006). This method ensures that the full range of parameters is sampled according to a LH design and within each sample the parameters are tested, one at a time. Initially, it takes $N$ LH sample points for $N$ intervals while varying each LH sample point $p$ times by changing each of the $n$ parameters one at a time, according to the OAT design (Van Griensven et al., 2006). Around each Latin Hypercube point a relative partial effect for each parameter is calculated. A final effect is calculated by averaging the partial effects over all $N$ LH points. Thus, local sensitivities (i.e. a partial effects) get integrated into a
global sensitivity measure. Having the same feature as the Monte Carlo sampling, i.e., a global screening method, LH sampling reduces the computational cost significantly ($n-1$ times), being more efficient (Van Griensven et al., 2006).

The method is very efficient, as the $N$ intervals in the LH method require a total of $N(p+1)$ evaluations. The relative importance of the parameters is determined by ranking the final effects from large to small (Van Griensven et al., 2006). Each relative importance can be divided by the sum of all relative importances to yield a normalized measure of relative importance.
We choose a step size that represents a fraction of 0.1 of the parameter search space. The twelve parameters selected for the sensitivity analysis are listed in Tables 1 and 2. The most sensitive parameters are selected such that the sum of their normalized relative importances reaches at least 95%.

After having selected the most important parameters by sensitivity analysis, the RAINLINK parameters are optimized with the method Standard Particle Swarm Optimization (SPSO-2011) (Clerc, 2012). Being a major improvement over previous PSO versions, with an adaptive random topology and rotational invariance, SPSO-2011 is a stochastic, effective, and efficient calibration method, as highlighted in recent studies with other hydrological and environmental models (Abdelaziz and Zambrano-Bigiarini, 2014; Bisselink et al., 2016; Pijl et al., 2018). The optimization is performed for the two RAINLINK sub-processes separately. First, the wet-dry classification parameters are calibrated, to make sure RAINLINK is able to correctly identify dry and rainy periods. Next, using the optimum parameters for the wet-dry classification, the rainfall retrieval parameters are calibrated. We have included all zero rainfall observations in the entire calibration process, both for the gauge-adjusted radar reference and for the RAINLINK estimates. Note that data from individual sub-links were used in the calibration process, so data from two links (in opposite directions) having the same link path were not averaged.

The goodness-of-fit measures chosen to drive the optimization and performance for the wet-dry classification and the rainfall retrieval processes are the Matthews Correlation Coefficient (MCC) (Matthews, 1975) and the modified Kling-Gupta efficiency (KGE) (Kling et al., 2012), respectively. Both are maximized towards an optimum value of 1. A 15-minute time interval from a given sub-link is considered dry if the reference is below 0.25 mm.

Due to the higher frequency of non-rainy 15-min intervals (data points), the process of wet-dry classification is considered an imbalanced classification problem. Employing recurrent metrics for binary classification, such as F1 score and Accuracy, may lead to inflated results. The Matthews Correlation Coefficient is less subjective and preferred since it informs how correlated the predictions and observations are, reaching a high score only if the prediction obtained good results in all the four confusion matrix categories (true positives ($TP$), false negatives ($FN$), true negatives ($TN$), and false positives ($FP$)) (Chicco and Jurman, 2020). The Matthews Correlation Coefficient is defined as

$$\text{MCC} = \frac{TP \cdot TN - FP \cdot FN}{\sqrt{(TP+FP)(TP+FN)(TN+FP)(TN+FN)}}. \tag{4}$$

The denominator is arbitrarily set to one when any of the four sums in the denominator is zero. Kling-Gupta efficiency is defined as

$$\text{KGE} = 1 - \sqrt{(\rho-1)^2 + (\beta-1)^2 + (\gamma-1)^2}, \tag{5}$$

with $\rho$ the Pearson correlation coefficient, $\beta$ the bias ratio

$$\beta = \frac{\mu_\text{e}}{\mu_\text{o}}, \tag{6}$$

and $\gamma$ the variability ratio

$$\gamma = \frac{\text{CV}_\text{e}}{\text{CV}_\text{o}} = \frac{\sigma_\text{e}\mu_\text{o}}{\mu_\text{e}\sigma_\text{o}}, \tag{7}$$

where $\mu$ and $\sigma$ are the mean and standard deviation of path-averaged rainfall intensity (mm h$^{-1}$) for CML estimates (e) and gauge-adjusted radar observations (o). CV is the coefficient of variation, defined as the ratio of the standard deviation and the mean.

### 2.4 RAINLINK validation

The validation was performed for both wet-dry and rainfall retrieval RAINLINK processes by using the newly calibrated parameters against its default parameters as given by Overeem et al. (2016a). In addition to MCC and following the confusion matrix, the assessment binary metrics, Accuracy, Sensitivity, and Specificity were computed as follows:

$$\text{Accuracy} = \frac{TP + TN}{TP + TN + FP + FN},\tag{8}$$

$$\text{Sensitivity} = \frac{TP}{TP + FN},\tag{9}$$

$$\text{Specificity} = \frac{TN}{FP + TN}.\tag{10}$$

As for the RAINLINK rainfall retrieval process, besides KGE and its components $\rho$, $\beta$ and $\gamma$, the CV of the residuals ($\text{CV}_{\text{res}}$), the percent bias (PBIAS) and root-mean-square error (RMSE) were employed.

$$\text{CV}_{\text{res}} = \frac{\sigma_{\text{res}}}{\mu_{\text{o}}},\tag{11}$$

$$\text{PBIAS} = 100 \frac{\sum_{i=1}^{n}(\text{e}_i - \text{o}_i)}{\sum_{i=1}^{n}\text{o}_i},\tag{12}$$

$$\text{RMSE} = \sqrt{\frac{\sum_{i=1}^{n}(\text{e}_i - \text{o}_i)^2}{n}}.\tag{13}$$

Finally, the level of agreement of daily rainfall patterns is analyzed graphically. RAINLINK's ability to estimate 15-minute path-average rainfall rates is also evaluated. Moreover, both agreement of accumulated rainfall for all individual CMLs and agreement of daily mean rainfall over the Netherlands estimated from the CML values (as time series) are considered, taking those links with over 75% of data availability into account.

## 3 Results and discussion

### 3.1 Calibration

#### 3.1.1 Wet-dry classification parameter optimization

The sensitivity analysis for the wet-dry classification process is performed at a 15-minute time interval. Table 3 provides the parameter ranking obtained considering the search space illustrated in Table 1. The most important parameters are, $\text{WD}_{\text{p2}}$,

**Table 3.** Wet-dry classification sensitivity analysis: $WD_{pn}$ (symbol) – wet-dry classification parameters, see description in Table 1. Note: [a] - Most sensitive parameters obtained from the Latin-Hypercube One-factor-At-a-Time analysis.

| Rank | Parameter (symbol) | Relative Importance (RI) | RI Normalized |
|---|---|---|---|
| 1[a] | $WD_{p2}$ | 77.68 | 0.48 |
| 2[a] | $WD_{p4}$ (median($\Delta P$)) | 35.07 | 0.21 |
| 3[a] | $WD_{p1}$ | 29.44 | 0.18 |
| 4[a] | $WD_{p5}$ (median($\Delta P_L$)) | 10.37 | 0.062 |
| 5[a] | $WD_{p3}$ (r) | 10.23 | 0.061 |
| 6 | $WD_{p7}$ | 2.16 | 0.013 |
| 7 | $WD_{p6}$ | 0.32 | 0.0019 |

$WD_{p4}$ (median($\Delta P$)), $WD_{p1}$, $WD_{p5}$ (median($\Delta P_L$)), and $WD_{p3}$ (r). The accumulated relative importance of these parameters

is 98%. The importance of the two thresholds ($WD_{p4}$ and $WD_{p5}$) was expected, because these parameters define the values for which an individual microwave link will be classified as rainy or not. However, the analysis performed here, which systematically evaluates all parameters together by maximizing a goodness-of-fit measure, reveals that the parameters $WD_{p2}$, $WD_{p1}$, and $WD_{p5}$ are important as well. The highest importance reached by the $WD_{p2}$ parameter highlights the rain-induced attenuation temporal correlation. Since, this parameter represents the number of previous hours over which the maximum value

of the minimum received power (max($P_{min}$)) is computed, it governs the wet-dry classification process by influencing on the attenuation (median($\Delta P$)) and specific attenuation (median($\Delta P_L$)) computation. It is important to highlight that the max($P_{min}$) is only computed if at least a minimum number of hours (defined by $WD_{p1}$) of data are available; otherwise it is not computed and no rainfall intensities are retrieved. The low ranking of the $WD_{p7}$ threshold is consistent with the findings of Overeem et al. (2016a), who report that including this step hardly changes results for a 12-day dataset when validating rainfall depths (i.e., the

total effect on the amounts, not the occurrence of wet and dry periods as such).

The five highest ranked parameters are now employed in the calibration, taking the ranges reported in Table 1 into account. Using particle swarm optimization (PSO), the parameters' dispersion and distributions across the search space have been computed for the 12-day calibration dataset (Fig. 2). The distributions are obtained for all solutions, the frequency histograms of the parameters are multi-modal and skewed (Fig. 2 (b)), reflecting the uncertainties in the optimum values.

The parameters $WD_{p1}$, $WD_{p2}$, $WD_{p3}$, $WD_{p4}$, and $WD_{p5}$ reach the optimum values equal to 7.5 h, 14.1 h, 19.7 km, $-2.7$ dB, and $-0.9$ db km$^{-1}$, respectively. Compared with the default values of these parameters, namely 6 h, 24 h, 15 km, $-1.4$ dB, and $-0.7$ dB km$^{-1}$, the difference is considered small for the parameters $WD_{p1}$, $WD_{p3}$, and $WD_{p5}$. However, the parameters $WD_{p2}$ and $WD_{p4}$ presented a reasonable difference compared to the default values. For those solutions with MCC value greater than 0.53, being classified as "behavioral" solutions (Zambrano-Bigiarini and Rojas, 2013), the median value of the parameters were

4.8 h, 10 h, 18.9 km, $-1.5$ dB, $-0.7$ dB km$^{-1}$ for $WD_{p1}$, $WD_{p2}$, $WD_{p3}$, $WD_{p4}$, and $WD_{p5}$, respectively. The values obtained for the calibrated parameters are based on the median of the "behavioral" solutions and are in line with the default parameters, except for $WD_{p2}$, which indicates a shorter period for computing the maximum of the minimum received power. A possible explanation for the optimized value of 10 h in contrast to the default value of 24 h is that this 10-h period is more representative

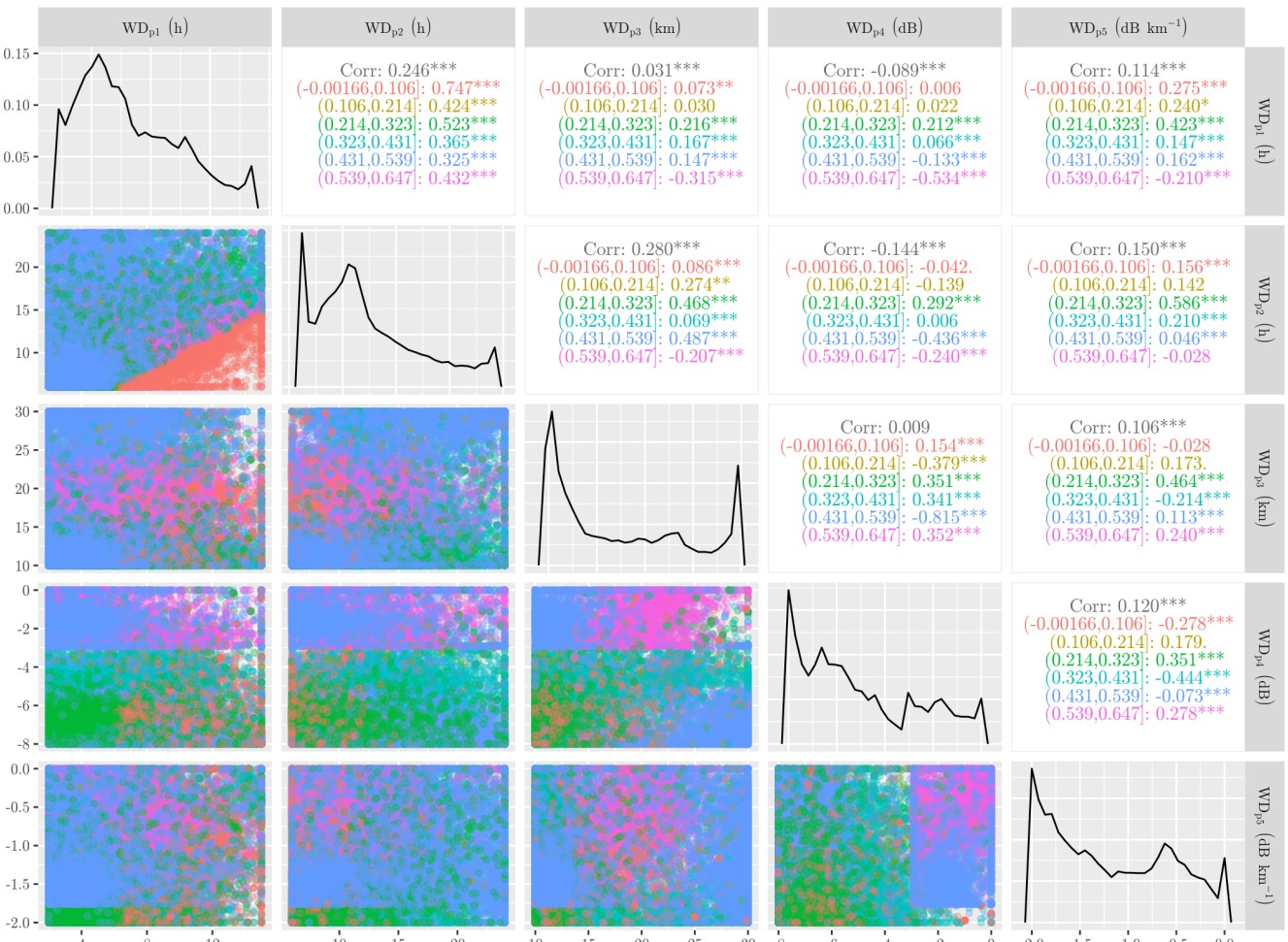

**Figure 2.** Calibration of the wet-dry classification parameters: panel showing the interaction between calibration parameters at different Matthews correlation coefficient (MCC) values. Note: the upper-right diagonal represents the correlation among the parameters, both for the total samples (black values) and for the grouped samples (color values) by MCC intervals. Significance levels: " *** " $p < 0.001$, " * " $p < 0.05$, " . " $p < 0.1$, and " no symbol " $p < 1$.

for the wet-dry classification. For instance, signal fluctuations such as due to changes in atmospheric moisture in the period 24 h to 10 h before the current interval, may result in wet-dry classification parameter values being less representative, which may lead to less accurate wet-dry classification.

It should be noted that, in spite of being gauge-adjusted, the radar product used here is not a perfect reference. Differences between radar sampling (indirect measurements aloft) and ground-based sensors can lead to significant errors (de Vos et al., 2019). Thus, accounting for this sampling difference could even further increase the value of the MCC metric. In particular for small rainfall events these errors can lead to false positive and false negative classifications.

The value of the WD$_{p1}$ parameter results in exclusion of 12% of the data points during the algorithm processing for both default and calibrated parameters sets (which have a similar value). This parameter has a direct relation with data availability, since it determines the minimum number of hours needed to compute max($P_{min}$). Note that max($P_{min}$) is only computed if at least a minimum number of hours of data are available; otherwise it is not computed and no rainfall intensities will be
retrieved (Overeem et al., 2016b). Although the calibration dataset has been selected considering rainy days, the number of non-rainy data points is much higher than the number of rainy data points, representing 93%, which is comparable to the average occurrence of dry spells in the Netherlands according to automatic weather stations. Thus, this calibration period can be considered representative for other periods within the same weather season.

### 3.1.2 Rainfall retrieval parameter optimization

The same sensitivity analysis and calibration are employed for the rainfall retrieval at the 15-minute time interval (Table 4), where zeroes in either CML and/or reference are also included. The sensitivity analysis presented here underlines the uncertainty associated with the microwave link measurements. The most sensitive parameters are the parameters RR$_{p5}$ ($\alpha$) and RR$_{p4}$ ($A_a$). The summed relative importance of these parameters is 95%.

The parameter RR$_{p4}$ is related to the correction of the attenuation due to wet antennas. This phenomenon is considered an
important source of extra attenuation and may cause significant rainfall overestimation if not sufficiently accounted for (Leijnse et al., 2008; Messer and Sendik, 2015; Overeem et al., 2016a).

Since the parameter RR$_{p5}$ represents a coefficient that determines the relative contributions of the minimum and maximum path-averaged rainfall intensities ($\overline{R}_{min}$ and $\overline{R}_{max}$, Eq. 3) to the 15-minute average rainfall intensity estimates, it is directly related to the temporal sampling strategy of the received signal power and has an important weight in the rainfall retrieval. In
a comparative study, de Vos et al. (2019) found that min/max sampling at a 15-minute time step (as employed by RAINLINK) outperforms instantaneous sampling in the Dutch climate. This underlines the importance of properly estimating RR$_{p5}$ ($\alpha$) for accurate rainfall retrievals.

The parameter RR$_{p3}$ ($F_t$) represents an outlier filter. Therefore, it seems reasonable to assume a threshold value based on expert judgment, because strict filtering would result in a high performance, but with a severe decline in the remaining number
of links. Using the default values of the parameters RR$_{p4}$ ($A_a$) and RR$_{p5}$ ($\alpha$) obtained in Overeem et al. (2013), Overeem et al. (2016a) applied a sensitivity analysis varying only the parameter RR$_{p3}$ ($F_t$), confirming that the default value equal to $-32.5$ dB km$^{-1}$ h$^{-1}$ (Overeem et al., 2013) is a reasonable trade-off between performance and retaining a significant number of links. Although considered unimportant by the sensitivity analysis in the range from $-100$ to $0$ dB km$^{-1}$ h, a proper calibration procedure is deemed important, the default value of RR$_{p3}$ ($F_t$) fixed at $-32.5$ dB km$^{-1}$ h$^{-1}$ is kept to prevent an excessive loss
of data. One way forward to calibrate RR$_{p3}$ ($F_t$) would be to include both the number of available links in the optimization or perform an optimization based on rainfall maps, which can be influenced by the underlying CML network density.

Figure 3 illustrates the interaction between parameters in the calibration procedure for the rainfall retrieval at different KGE-values. This figure shows that the regions with the highest KGE-values (greens and blue points) correspond mainly to values ranging from 1 to 2.5 dB for RR$_{p4}$ and from 0.17 to 0.30 for RR$_{p5}$. We classified the solutions greater than 0.45 as

**Table 4.** Rainfall retrieval sensitivity analysis: $RR_{pn}$ – rainfall retrieval parameters, see Table 2.

| Rank | Parameter (symbol) | Relative Importance (RI) | RI Normalized |
|---|---|---|---|
| 1 | $RR_{p5}$ ($\alpha$) | 1071.18 | 0.84 |
| 2 | $RR_{p4}$ ($A_a$) | 143.92 | 0.11 |
| 3 | $RR_{p1}$ | 27.96 | 0.02 |
| 4 | $RR_{p2}$ | 17.39 | 0.01 |
| 5 | $RR_{p3}$ ($F_t$) | 8.32 | 0.006 |

(a)  (b)

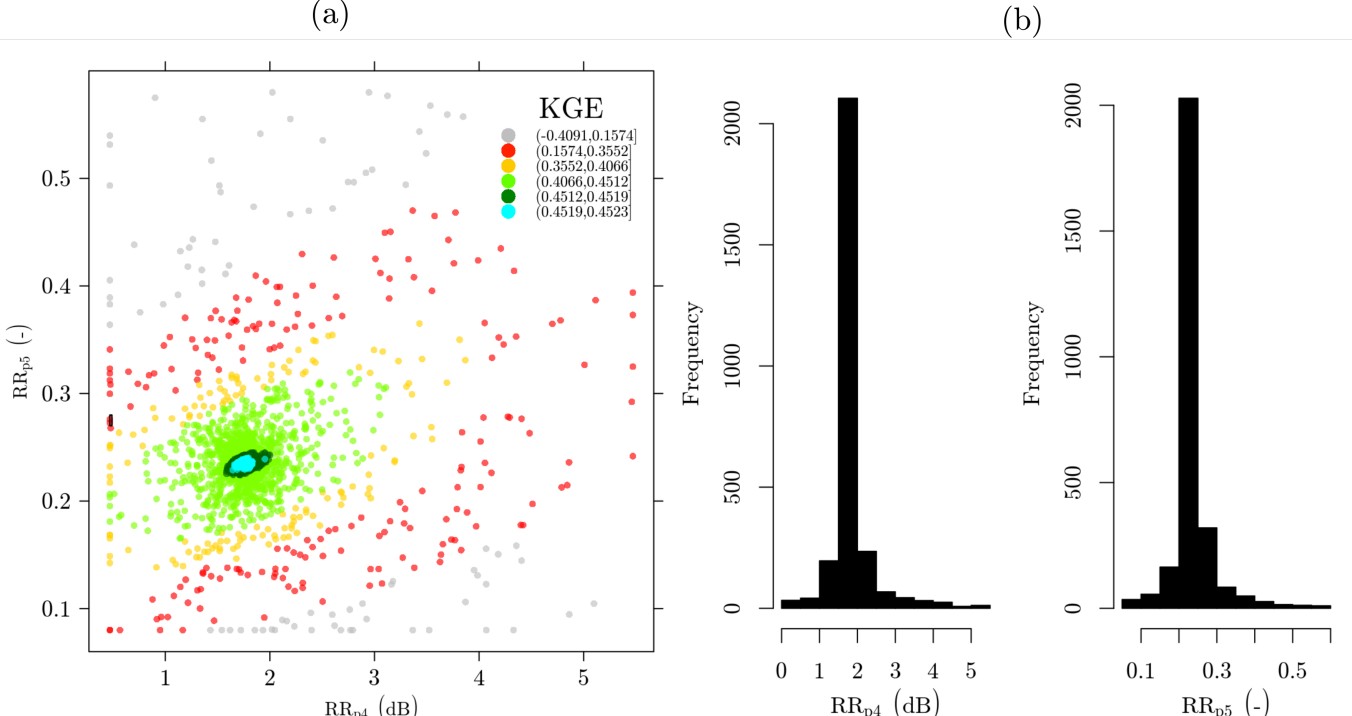

**Figure 3.** Rainfall retrieval performance projected onto the parameter space: dotty plot showing the interaction between calibration parameters at different Kling-Gupta efficiency (KGE) values.

"behavioral" solutions (dark green and blue points in Fig. 3 (a)). Different from the calibration of the wet-dry classification process, we observe a distribution of the parameters less skewed and with a well defined mode (Fig. 3 (b)). Thus, for the respective parameters $RR_{p4}$ and $RR_{p5}$, the optimum values, 1.7 and 0.23, are almost identical to the median values for the "behavioral" solutions, 1.74 and 0.24.

The parameter $RR_{p4}$ shows a more pronounced dispersion than the parameter $RR_{p5}$. $RR_{p4}$ is related to wet antenna attenuation and varies depending on the ambient conditions, e.g. while there is dew, rain water or melting precipitation (the latter unlikely in this study) present on the antenna covers (Leijnse et al., 2008; Overeem et al., 2016b; Uijlenhoet et al., 2018). It may also vary depending on the type of antenna cover. Finally, in the rainfall retrieval algorithm it is always assumed that,

whenever it rains, both antennas of a microwave link are wet, whereas in reality none or only one antenna may be wet. Hence, it is unlikely that all CMLs across the considered study area will share the same excess signal attenuation in terms of magnitude, timing and spatial occurrence. In principle, each single CML is expected to have its own time-varying set of values of the parameter $RR_{p4}$. This implies great uncertainty in the overall optimum value for the time period and region of interest. These parameters are expected to be positively correlated. Likely, higher $RR_{p5}$ values lead to higher rain intensities, increasing the weight of the maximum attenuation and consequently a higher value of $RR_{p4}$ would become necessary to compensate for the extra attenuation, decreasing the rain intensity estimates.

It is apparent from Fig. 3 that the parameter $RR_{p5}$ reaches its optimum value at 0.23, which is much lower than RAINLINK's default value of 0.33. This implies that the maximum and minimum path-averaged rainfall intensities ($\overline{R}_{max}$, $\overline{R}_{min}$) have respective weights of 0.23 and 0.77 in the computation of the best estimate of the 15-minute mean path-averaged rainfall intensity. However, a smaller spread around the optimum value compared to the other parameters can be observed, indicating a moderate uncertainty around the optimum. Note that the value of $\alpha$ is related to the temporal distribution of path-average rainfall intensities within 15-minute intervals, which is influenced by the lengths of the links as well as by the rainfall space-time variability. This suggests that the optimum parameter value will depend on both link network topology and rainfall climatology.

Its important to highlight that we did not calibrate the power-law coefficients. Since they are physically-based, we used values obtained in dedicated experiments representative for the Dutch climate (Leijnse, 2007). For other countries, the International Telecommunication Union (ITU) presents recommendations (International Telecommunication Union, 2005). However, these are not representative for all climates. A physically-based approach which derives these coefficients from drop size distribution observations and scattering computations is preferred compared to optimizing these coefficients in a statistical manner, especially for frequencies higher than 35 GHz. The drop size distribution dependence of the k-R relation in the frequency range of approximately 20-35 GHz is considered small compared to errors from wet antenna attenuation or erroneous wet-dry classification. Although a physically-based approach is considered better, a calibration of power-law coefficients may be a way forward for regions which lack disdrometer data (Ostrometzky and Messer, 2020).

### 3.2 Validation

After the parameter optimization using the 12-day calibration dataset from 2011, the optimized and default parameter sets are applied to a 3-month validation dataset from July, August, and September 2012. The 15-minute path-average rainfall estimates were aggregated to hourly and daily path-average rainfall estimates if CML-availability was at least 75%, resulting in data from on average 2,783 sub-links for both the default and optimized parameters. Thus, given that after the RAINLINK pre-processing on average 2,800 sub-links are left, data availability reduces by approximately 0.7% for both default and optimized parameters due to the pre-processing.

#### 3.2.1 Wet-dry classification validation

Figure 4 highlights that the wet-dry classification process by using calibrated parameters performs better in terms of MCC and Accuracy metrics, 0.40 and 0.96 against 0.37 and 0.95 for the default parameters, respectively. However, the Sensitivity

metric shows that the calibrated parameters are worse for classifying the true positive rate, the default parameters set reach a value of 0.51 against 0.49 for the calibrated parameters set. We find a MCC value of 0.4 for the validation dataset, being smaller than the MCC threshold for "behavioral" solutions, i.e., 0.53. This occurred because the optimization might not have generalized well enough the wet-dry classification process. It was focused on the calibration dataset, capturing many details and noise, and subsequently failed to capture a different trend from another dataset, i.e., became an overfitted model. Thus, the performance for the validation dataset was worse, because the calibration dataset will not be entirely representative for other periods. A solution could be to increase the size of the calibration dataset, encompassing more characteristics and trends about the phenomenon.

In spite of having the same Specificity value, we can observe in confusion matrices (Fig. 4, green cells for line and column 1) that the calibrated parameters set classified more dry periods rightly than the default parameters set. Thus, considering the MCC feature, which aims to evaluate all elements of the confusion matrix (false positive, false negative, true positive, and true negative), the calibrated parameters outperform the default ones. Approximately 50% of the rainy events are classified as dry (i.e., false negatives), both for the calibrated and default parameter sets. Using a convolutional neural networks for classifying wet-dry periods, Polz et al. (2020) found a proportion of 25% for false negatives, approximately.

According to the wet-dry observations of the reference during the validation period, we observed that 97% of the data points represent non-rainy intervals. Being just four percentage points higher than the calibration period (93%), the fraction of dry periods can be considered comparable to each other. However, the fraction of rainy periods for the calibration period (7%) is more than twice as high as for the validation period (3%). This implies that the calibration dataset is at least different with respect to the validation dataset concerning the percentage of rainy periods, which may have resulted in a lower MCC value for validation.

### 3.2.2 Rainfall retrieval validation

Figure 5 illustrates the performance in terms of daily path-average rainfall estimates for the two tested parameter sets, i.e., calibrated and default. In general, the metrics for the calibrated parameters are slightly better than those for the default parameters. The values improve from 0.37 to 0.45 for KGE, from 6.37 mm to 5.75 mm for RMSE, from 2.5 to 2.27 for $CV_{res}$, and from 0.42 to 0.46 for $\rho$.

The main improvement is observed for the percent bias (PBIAS). Even if both parameter sets lead to overestimates compared to the reference, the rainfall depth retrieved when using the calibrated parameters shows 10.6 percentage points less overestimation compared to using the default parameters. In addition to $\rho$, the bias ratio ($\beta$) and the variability ratio ($\gamma$) are incorporated into the KGE metric (Equations (5)–(7)). For the default parameters $\beta$ and $\gamma$ are 1.24 and 0.99, respectively. For the calibrated parameters the values of $\beta$ and $\gamma$ are 1.13 and 0.99, respectively. All of the three KGE components have their ideal value at unity and the higher value of KGE when using the calibrated parameters is due to a better bias performance. Overall, the calibrated parameters outperform the default parameters.

Next, the performance of 15-minute path-averaged rainfall estimates is investigated. Table 5 summarizes RAINLINK's performance when the default and calibrated parameters are applied for different rainfall thresholds. The calibrated parameter

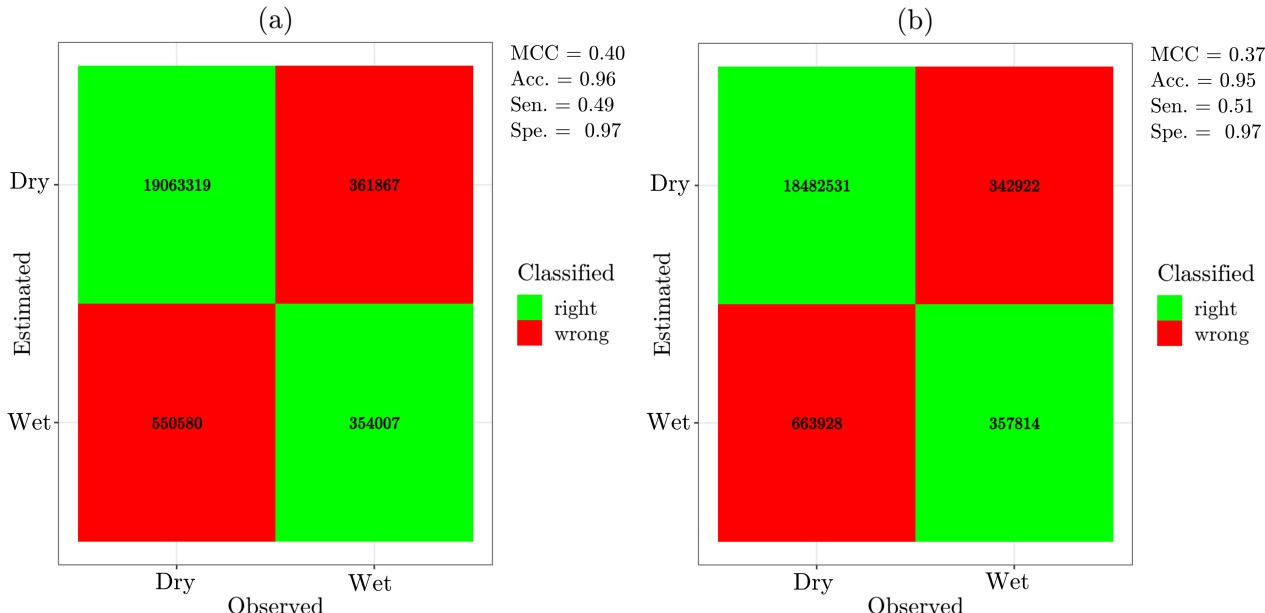

**Figure 4.** Confusion matrices and binary classification metrics for the wet-dry classification process: (a) results of the wet-dry classification using calibrated parameters and (b) results of the wet-dry classification using default parameters. Note: Matthews correlation coefficient (MCC), Accuracy (Acc.), Sensitivity (Sen.), and Specificity (Spe.) metrics.

375    set yields a better performance of RAINLINK in terms of KGE, RMSE, and $CV_{res}$ for all thresholds. As for PBIAS, the default parameters outperform the calibrated ones for the thresholds "Reference > 0" and "Reference > 1", whereas the calibrated parameters show better performance for the remaining thresholds. One can also observe that, if a threshold is only applied to the reference and consequently the false positives are removed, RAINLINK shows a large underestimation with respect to the reference. This underestimation is not observed if either RAINLINK or the reference are above the threshold. This

380    indicates that the observed underestimation is due to RAINLINK estimating zero rain when the reference suggests that it is raining. This may be related to differences in spatial and temporal sampling, although we are not able to provide a conclusive explanation. Indeed, the false positives presented a strong effect on PBIAS, because when they were removed (i.e., "Reference > 0"), PBIAS changed significantly. As for Polz et al. (2020) a different behavior was observed, maybe due to a lower number of false positives or due to a different distribution of false positives.

385    The $\rho$ goodness-of-fit metric results in a better performance for the default parameters, where the CML rainfall estimates are used in the thresholds. On the other hand, when just the radar reference is considered in the thresholds, the calibrated parameters set reach better $\rho$ performance.

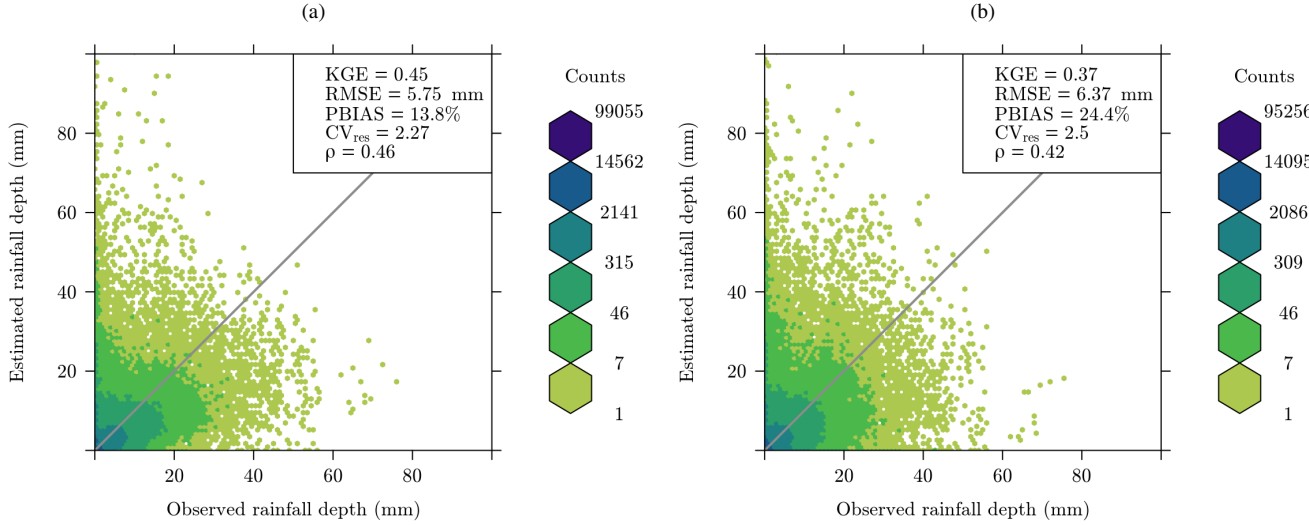

**Figure 5.** Daily path-averaged rainfall depth comparison of CML rainfall estimates against gauge-adjusted radar data: (a) rainfall retrieved using calibrated parameters; (b) rainfall retrieved using default parameters.

**Table 5.** 15-minute path-averaged rainfall depth performance for different thresholds. Note: Reference is the gauge-adjusted radar data.

| Thresholds of rainfall (mm) | RAINLINK with parameters | KGE | RMSE (mm) | PBIAS (%) | $CV_{res}$ | $\rho$ |
|---|---|---|---|---|---|---|
| Reference OR RAINLINK > 0 | Default | 0.21 | 1.06 | 33.10 | 2.69 | 0.28 |
| | Calibrated | 0.27 | 0.99 | 8.50 | 2.23 | 0.27 |
| Reference OR RAINLINK > 0.1 | Default | 0.18 | 1.15 | 31.60 | 2.49 | 0.26 |
| | Calibrated | 0.24 | 1.05 | 7.40 | 2.10 | 0.25 |
| Reference OR RAINLINK > 1 | Default | −0.16 | 2.20 | 56.20 | 1.95 | 0.00 |
| | Calibrated | −0.05 | 2.02 | 22.60 | 1.68 | −0.02 |
| Reference > 0 | Default | −0.11 | 1.01 | −38.00 | 1.30 | 0.48 |
| | Calibrated | 0.03 | 0.92 | −42.20 | 1.16 | 0.50 |
| Reference > 1 | Default | −0.46 | 1.92 | −39.70 | 0.89 | 0.38 |
| | Calibrated | −0.31 | 1.77 | −45.20 | 0.78 | 0.40 |
| No threshold (zero included) | Default | 0.33 | 0.28 | 33.10 | 10.55 | 0.42 |
| | Calibrated | 0.42 | 0.25 | 18.50 | 9.37 | 0.45 |

When no thresholding is applied the calibrated parameters clearly perform better than the default ones in terms of KGE and PBIAS values. With respect to data availability, the calibrated and default parameter sets contain 15.6% and 12.3% less observations after running all of RAINLINK's processing steps than the entire data set, respectively.

390

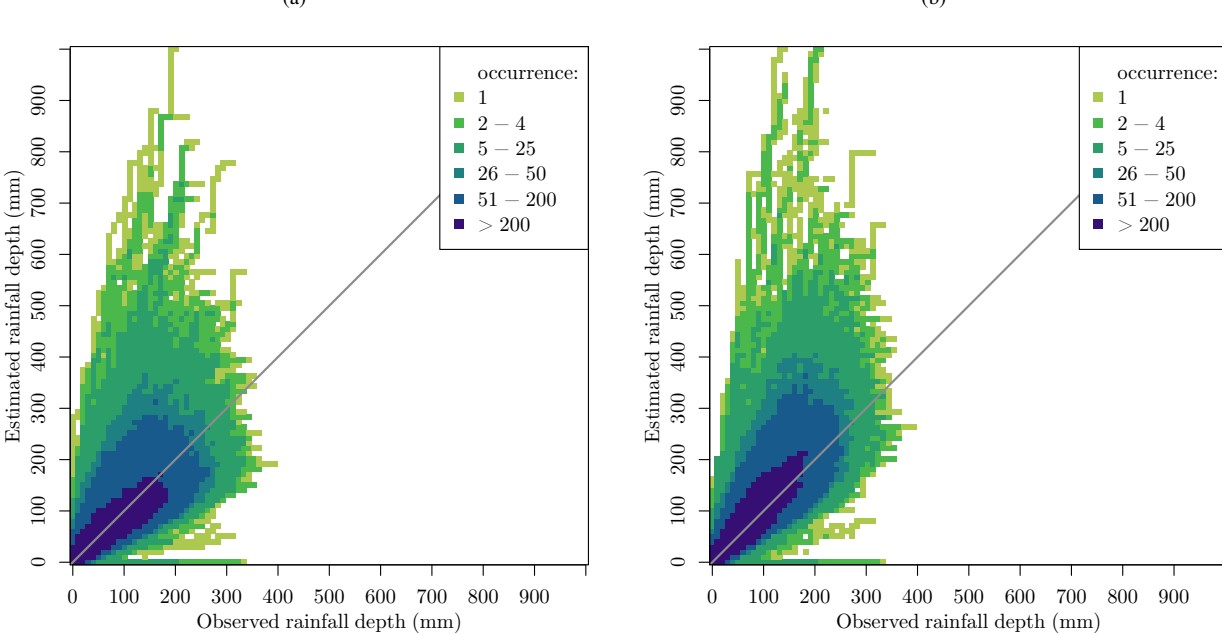

**Figure 6.** Density plots of the double mass curves of all individual CMLs with respect to the gauge-adjusted radar reference at 15-minute time intervals: (a) rainfall accumulations retrieved by RAINLINK with calibrated parameters; (b) rainfall accumulations retrieved by RAINLINK with default parameters.

Reevaluating the Overeem et al. (2016b) study employing default parameter values, de Vos et al. (2019) find 5.75%, 2.84, and 0.27 for PBIAS, CV$_{\mathrm{res}}$, and $\rho$, respectively, for path-average 15-minute rainfall depths, and for link or radar larger than 0 mm. Differences with respect to the performance obtained here for the default parameter values (33.10%, 2.69, and 0.28 for PBIAS, CV$_{\mathrm{res}}$, and $\rho$, respectively) can be explained by the fact that the underlying data for both studies are from different periods, with different durations ($\sim$ 20 months for the months of February-October in de Vos et al. (2019) and 3 months for the months of June-August here). Possibly, the wet-dry classification using default parameters applied by de Vos et al. (2019) results in less false positives or due to the longer period the false negatives compensate for the false positives, resulting in a lower PBIAS value. The summer of 2012 was rainy, with 286 mm of rain compared to the climatological average of 225 mm, averaged over the Netherlands. For the central weather station in the Netherlands, a long precipitation duration of 153 hours was observed compared to the climatological average of 121 hours over the summer months June, July, and August. This could be a reason for differences in PBIAS, although this summer is also part of the 613-day dataset evaluated in de Vos et al. (2019).

Figure 6 shows density plots for all CML double-mass curves, i.e., the relation between the accumulations of rainfall retrieved by RAINLINK and that obtained from the gauge-adjusted radar reference. This figure shows that the class with the highest occurrence coincides with the diagonal, indicating a reasonable agreement between the estimates and the observations. A considerable dispersion above the diagonal is found for both the calibrated and the default parameters. However, it is clear

that with the calibrated parameters, this dispersion is less severe. This overestimation observed in the double-mass curves is in line with the PBIAS values reported earlier (Table 5), being caused by the higher presence of false positive observations. Identifying the extra attenuation as the main source of error, de Vos et al. (2019) report a similar behavior of the double-mass curves for instantaneous signal power sampling, although the considered period and hence the meteorological circumstances are partly different.

So far small improvements in the rainfall retrievals are obtained when employing the calibrated parameters through the stochastic method Particle Swarm Optimization (PSO). However, analyzing the average over an area, in this case the Netherlands, more substantial improvements are found. Figure 7 shows time series of the daily mean rainfall depth over the Netherlands, i.e., for each day the mean of all CML rainfall estimates is computed.

By employing the calibrated parameters, all metrics improve with respect to the default parameters. The values of KGE, RMSE, PBIAS, $CV_{res}$, and $\rho$ improve from 0.49, 3.39 mm, 24.4%, 1.32, and 0.59 to 0.57, 3.07 mm, 13.8%, 1.21, and 0.63, respectively. Since the CML rainfall estimates are averaged over a $\sim$35,500 km$^2$ area not taking into account how they are distributed, the PBIAS and $\beta$ values stay the same (Fig. 5). On the other hand, the variability and similarity (correlation), expressed by KGE components $\gamma$ and $\rho$, respectively, are slightly better. In spite of not being a homogeneous network, the CMLs are observing in the entire Netherlands, having a high enough spatial representativity for computing a spatial average rainfall. Thus, for the areal time series obtained by employing calibrated parameters, $\gamma$ and $\rho$ are equal to 0.83 (0.81 for default) and 0.63 (0.59 for default), respectively. The $\gamma$ value closer to unity, confirms that the estimated rainfall time series vary to the same extent as the observed rainfall time series. Hence, as concluded from the path-averaged rainfall evaluation, the main improvement provided by the calibrated parameters as compared to the default parameters is a lower relative bias.

For both sets of parameters, calibrated and default, CML-derived rainfall estimates correspond reasonably well to the gauge-adjusted radar rainfall estimates. For path-averaged daily rainfall an improvement is found when calibrated parameter values are employed, especially in terms of relative bias. Results further improve when rainfall estimates are averaged over the entire Netherlands. Differences in calibrated parameter values with respect to the default ones may be caused by the calibration being performed over different events in June and July 2009 and in 2011 for the default parameters (Overeem et al., 2011, 2013). Moreover, the calibration here is done with a state-of-the-art and efficient method.

### 3.3 Search space of parameters

For some parameters in Tables 1 and 2 a wider search space could have been chosen. For $WD_{p2}$ and $RR_{p2}$ a maximum value of 24 h was chosen, implying that data from the previous day are needed. Because the calibration dataset is not continuous, it was not feasible to use a larger value for $WD_{p2}$. In both cases 24 h seems reasonably long for a reliable computation. The maximum allowed value of 24 h for $RR_{p2}$ may even be beneficial for the reference level determination. If this value would become longer than 24 h, varying meteorological conditions (e.g. related to changes in relative humidity) may affect the accuracy of the reference level determination, being less representative of the reference level just before a rainfall event. For the radius $WD_{p3}$ the minimum value is 15 km. A lower value could be tested, but given the network density (Fig. 1), this is expected to lead to a (severe) reduction of available sub-links. This is because the wet-dry classification needs a minimum

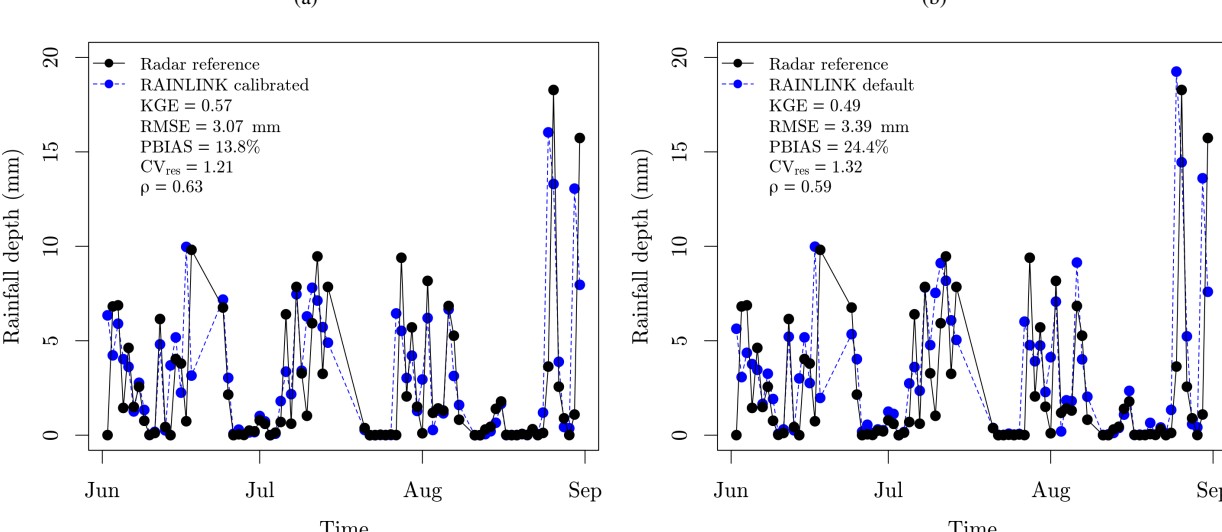

**Figure 7.** Comparison of the daily mean rainfall depth time series for the entire Netherlands during the summer months June, July and August 2012: (a) rainfall time series retrieved by RAINLINK with calibrated parameters; (b) rainfall time series retrieved by RAINLINK with default parameters.

440 number of nearby links, which is more difficult to achieve in case of a smaller radius. The employed minimum value for $WD_{p6}$ is already quite low. The wet-dry classification is expected to become more reliable when more sub-links are involved. Hence, it does not seem sensible to choose an even lower minimum value.

## 4   Conclusions

A novel and reliable method for the objective estimation of optimum parameter sets for RAINLINK and potentially for 445 other CML-based rainfall retrieval algorithms has been presented and tested. Using a 12-day dataset, the calibration was performed by means of a stochastic approach, Particle Swarm Optimization (PSO), preceded by a sensitivity analysis selecting the parameters to be optimized. The optimized parameters were determined according to optimum goodness-of-fit values and for the median of "behavioral" solutions, i.e., those solutions performing better than a threshold. Table 6 summarizes the values of RAINLINK's optimized parameters and the default ones.

450 The validation of daily path-averaged CML rainfall estimates over three summer months reveals a reasonable improvement for the calibrated parameters compared to the default values. When daily path-averaged values are averaged over the entire surface area of the Netherlands, the improvement becomes much stronger. The aggregation over an area tends to limit the effects of representativeness errors in the rainfall estimates and yields information with an acceptable performance for hydrological and meteorological applications. This result is important, because from a general perspective, hydrological and meteorological 455 scales of application are defined over areas, e.g., watersheds, climate zones, political and administrative regions, etc. Com-

**Table 6.** RAINLINK parameters: default and calibrated values (median of "behavioral" solutions). Note: calibrated parameter values are in bold font.

| Parameter description | Symbol and unit | Default value | Calibrated value |
|---|---|---|---|
| $WD_{p1}$ – Minimum number of hours needed to compute $\max(P_{min})$ | – (h) | 6 | **4.8** |
| $WD_{p2}$ – Number of previous hours over which $\max(P_{min})$ is to be computed (also determines period over which cumulative difference $F$ of outlier filter is computed) | – (h) | 24 | **10** |
| $WD_{p3}$ – Radius | $r$ (km) | 15 | **18.9** |
| $WD_{p4}$ – Attenuation threshold | $\mathrm{median}(\Delta P)$ (dB) | −1.4 | **−1.5** |
| $WD_{p5}$ – Specific attenuation threshold | $\mathrm{median}(\Delta P_L)$ (dB km$^{-1}$) | −0.7 | **−0.7** |
| $WD_{p6}$ – Minimum number of available (surrounding) links | – (–) | 3 | 3 |
| $WD_{p7}$ – Minimum received power threshold | – (dB) | 2 | 2 |
| $RR_{p1}$ – Minimum number of hours that should be dry in preceding period | – (h) | 2.5 | 2.5 |
| $RR_{p2}$ – Period over which reference level is to be determined | – (h) | 24 | 24 |
| $RR_{p3}$ – Outlier filter threshold | $F_t$ (dB km$^{-1}$ h) | −32.5 | −32.5 |
| $RR_{p4}$ – Wet antenna attenuation | $A_a$ (dB) | 2.3 | **1.74** |
| $RR_{p5}$ – Temporal rainrate distribution coefficient | $\alpha$ (–) | 0.33 | **0.24** |

pelling improvements were achieved not only in terms of the performance of CML rainfall estimates as such, but also with respect to the choice of parameter values, which are now underpinned in a more objective way.

In fact, we now have a way to analyze the sensitivity and optimize stochastically all parameters used in a rainfall retrieval algorithm. The proposed methodology is applicable for different CML networks, climates, and algorithms, where either rain gauge or (gauge-adjusted) radar data can be used as reference. In case of other sampling strategies than min/max the algorithm can be easily adapted. Ideally, optimized parameters would be obtained for different seasons. Hence, for each processing period a dedicated parameter set would be obtained.

Fencl et al. (2019) underline the importance of considering the rainfall properties in the quantification of wet antenna attenuation, where a fixed value may lead to overestimation of heavy rainfalls. This can lead to an increase in the computational cost however, especially in case of extensive CML datatsets. We also recommend to extend our algorithm by adding an extra goodness-of-fit criterion to the optimization regarding the sub-link data availability after running RAINLINK's processing steps (de Vos et al., 2019). This could lead to improved coverage of CML rainfall estimates. In general, quantifying the effect of processing steps on data availability is important. Moreover, due to the large impact of false positives on PBIAS, a calibration of the rainfall retrieval process taking into account the wet-dry classification from the reference should be considered for further

research. Thus, an overestimation of wet antenna attenuation that has to compensate for the long-term rainfall overestimation from false positives would be avoided.

As a recommendation, studies could be conducted by testing the convergence and performance of different goodness-of-fit measures in addition to the Kling-Gupta efficiency (Kling et al., 2012). Moreover, one could optimize the parameters using rain gauges near the CMLs as reference in order to exclude deviations that are sometimes found in radar rainfall observations. Moreover, representativeness errors between radars, measuring aloft, and CMLs, measuring near the Earth's surface, can affect comparisons between the two. This especially holds for short time intervals, as short as 15 minutes in this study.

In spite of having stochastic properties and aiming to explore the uncertainties affecting rainfall retrievals from CMLs, the approach proposed here is not a panacea. In regions without reliable rainfall ground-truth the calibration of rainfall retrieval algorithm parameters can be a challenge (Chwala and Kunstmann, 2019). Hence, we recommend the set-up of experiments in regions with little ground-based rainfall information in order to optimize parameters for specific networks and climates, or even to improve rainfall retrieval algorithms such as RAINLINK themselves. As an alternative, parameters could be optimized in a well-gauged region having a similar climate and CML network as the ungauged region for which CML rainfall estimates are desired.

Further research can be conducted to test how the parameter range affects the importance of parameters in this approach. Specifically, even wider parameter ranges could be tested. Moreover, a longer calibration period could be analyzed to make the optimized parameters more generally applicable to other data from other periods. This especially holds for the wet-dry classification process.

Comparing CML-derived rainfall maps and gauge-adjusted radar observations, Overeem et al. (2016b) found a better performance for the summer season than for the winter season in the Netherlands, among others due to the absence of snow and melting precipitation. The rainfall type during the Dutch summer is largely of a convective nature, bearing some resemblance with that in regions characterized by (sub)tropical climates, which often lack surface rainfall observations. As a consequence, we believe CML rainfall monitoring is especially promising for low- to middle-income countries, typically having (sub)tropical climates.

*Acknowledgements.* We gratefully acknowledge Ronald Kloeg and Ralph Koppelaar from T-Mobile NL for providing the commercial microwave link data. This study was financially supported by the São Paulo Research Foundation (FAPESP) under the grant number 2017/09708-7. We thank the two anonymous reviewers for their constructive comments.

*Data availability.* The CML data are available via https://doi.org/10.4121/uuid:323587ea-82b7-4cff-b123-c660424345e5. The gauge-adjusted radar data can be obtained at https://dataplatform.knmi.nl/dataset/rad-nl25-rac-mfbs-5min-2-0.

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
