# Peer review of "Rainfall retrieval algorithm for commercial microwave links: stochastic calibration"

_Atmospheric Measurement Techniques, 2021_

## Author Response (AR1)

**Author's reply to the referees comments to manuscript AMT-2021-34**

**Anonymous referee 1**

**Please see the file with track changes to check the lines edited.**

The original referee's comments are written in bold and the author's replies are written in regular font.

**Review summary for "Rainfall retrieval algorithm for commercial microwave links: stochastic calibration"**

**In this paper, the authors address a known problem: many different algorithms and approaches to retrieve the rainfall via commercial microwave links have been presented in the past. However, most of the presented model-based approaches are sensitive to various design parameters of the specific algorithm. The authors return to a previously published algorithm - the RAINLINK, describe the problematic-sensitivity to specific design parameters, and suggest a methodology to pinpoint the most important parameters, and to better calibrate these parameters (which the authors did previously via empirical calibration).**

We gratefully thank the Referee for the constructive comments and recommendations.

**The problem at hand is indeed important, and the results presented by the authors are encouraging. However, in my opinion there are two major issues that should be resolved prior to the publication of this paper:**

**1.    Focusing on model-based only approaches is limited. Once training data is available, and stochastic models are considered, many current deep-learning algorithms can be implemented, which can potentially solve the parameter-calibration problem by suggesting a data-driven solution. E.g., see [1], among others. Thus, the solution presented by the authors here should be compared to such updated tools, or at least be discussed regarding the disadvantages and advantages between the presented approach and such data-driven approaches.**

Reply:

We appreciate this comment. Data-driven approaches indeed deserve further investigation (as in Pudashine et al., 2020), but are beyond the scope of our manuscript (a perspective referee 2 seems to agree with, according to his/her introductory statement). We think that a data-driven solution is not feasible for places or countries without sufficient reference data to train the model, such as a gauge-adjusted radar dataset which provides full coverage over a CML network. Although a data-driven solution could be used in our study, we focus on a stochastic calibration approach, which we expect to be more widely applicable since it does not require a large training dataset. This is especially important for the low- and middle-income countries in (sub)tropical regions, which would benefit most from the complementary rainfall information CMLs can deliver, and which often lack extensive reference datasets.

We added the following sentences:

L60: "...Machine learning supervised algorithms have been used for rainfall retrieval via CMLs, improving the performance of this kind of rainfall measurement (Pudashine et al., 2020; Habi and Messer, 2021). Although representing a recent advancement, data-driven solutions are not feasible for places or countries without sufficient reference data to train the machine learning algorithms, such as a gauge-adjusted radar dataset which provides full coverage over a CML network...".

**2.     The authors emphasize that their approach gives at least a partial solution for different climate regions. However, in these cases, it is important to consider some physical parameters that might affect the accuracy of the outcome, such as the power-law coefficients themselves. Specifically, these parameters are climate-sensitive, as was presented in past studies. How the implementation of such parameters into the calibration scheme affect the results?**

Reply:

Thanks for the suggestion, which definitely helps to improve this paper. As the power law parameters are physically-based, we used values obtained in dedicated experiments representative for the Dutch climate (Leijnse, 2007, p. 65). For other countries, the International Telecommunication Union (ITU) presents recommendations (ITU-R Recommendation P.838-3), but these are not representative

for all climates. In our opinion, a physically-based approach which derives these coefficients from drop size distribution observations and scattering computations is preferred compared to optimizing these coefficients in a statistical manner. We added this to the discussion and also mention that taking these parameters in the optimization into account may be a way forward for regions which lack disdrometer data. For such regions, an alternative approach would be to use disdrometer data from a similar climate as for the CML data.

At the same time, the importance of the power-law coefficients and their estimation through a physically-based approach should not be overrated either. Since the value of the exponent is close to 1, other parameters can compensate for the values of the power-law coefficients as long as the non-linearities are not too large. Moreover, previous studies show the exponent is quite invariant to the shape of the drop size distribution.

We wrote:

L329: "…Its important to highlight that we did not calibrate the power-law coefficients. Since they are physically-based, we used values obtained in dedicated experiments representative for the Dutch climate (Leijnse et al., 2007). For other countries, the International Telecommunication Union (ITU) presents recommendations (International Telecommunication Union, 2005). However, these are not representative for all climates. A physically-based approach which derives these coefficients from drop size distribution observations and scattering computations is preferred compared to optimizing these coefficients in a statistical manner. However, taking these parameters in the optimization into account may be a way forward for regions which lack disdrometer data…"

**All in all, this paper provides an interesting approach, and is well written. However, it should relate also to recent advancement in this field that address the same general problem via machine learning tools.**

We appreciate your feedback and we've added recent machine learning tools to the manuscript introduction.

**[1] H. V. Habi and H. Messer, "Recurrent Neural Network for Rain Estimation Using Commercial Microwave Links," in *IEEE Transactions on Geoscience and Remote Sensing*, doi: 10.1109/TGRS.2020.3010305.**

Leijnse, H., Uijlenhoet, R., and Stricker, J. N. M.: Rainfall measurement using radio links from cellular communication networks, Water Resour. Res., 43, WR005631, https://doi.org/10.1029/2006WR005631, 2007.

Pudashine, J., A. Guyot, F. Petitjean, V.R.N. Pauwels, R. Uijlenhoet, A. Seed, M. Prakash, and J.P. Walker, 2020: Deep Learning for an improved prediction of rainfall retrievals from commercial microwave links. *Water Resour. Res.*, **56**, e2019WR026255, doi:10.1029/2019WR026255.

**Author's reply to the referees comments to manuscript AMT-2021-34**

**Anonymous referee 2**

**Please see the file with track changes to check the lines edited.**

The original referee's comments are written in bold and the author's reply are written in regular font.

**Review summary for "Rainfall retrieval algorithm for commercial microwave links: stochastic calibration"**

**Summary:**

**This manuscript presents a new stochastic calibration of the most important parameters of the well established RAINLINK method, which is used for processing CML attenuation data to derive rainfall estimates. Since the RAINLINK method is applied by an increasing number of researchers, a detailed sensitivity analysis and an improved calibration would be an important contribution that could provide guidance for choosing the RAINLINK parameters in future analyses. The manuscript is well written and well structure and would be of interest for readers of AMT. I found several major issues with the analysis, though. Solving these issues will require to redo most parts of the analysis. Hence, I recommend a major revision. I do, however, not see the need to add a comparison of RAINLINK with other methods to this manuscript. The focus on calibration and sensitive analysis of parameters is a reasonable scope for one manuscript.**

We gratefully thank the Referee for the constructive comments and recommendations.

**General comments and recommendations:**

**1. Short calibration period with potentially biased fraction of wet and dry periods:**

**The calibration period is fairly short, only 12 days, and hence might not cover challenging dry periods with strong fluctuations, noise or artifacts. Since these 12 days have been selected from a longer period, I assume that these are all**

**rainy days. If this is the case, this would shift the false-positive and false-negative rates in the validation period compared to the calibration period. As a results the optimal wet-dry parameters from the calibration period might not be optimal for the validation period (see my comment on L104). This can lead to unexpectedly high numbers of false classifications. Based on the result in table 5, I conclude that this is the case here. According to my interpretation, a large number of false-positives contributes to the overall CML rainfall sum, see my comment on L311 for more details. I strongly recommend to, either chose calibration and validation data so that the wet-dry rations is similar, or to use a performance metric that is more robust to changes in this ratio.**

Reply:

We appreciate this comment that would make the paper more complete when implemented. In short: we accepted this suggestion and employed a more robust performance metric.

Note that this dataset was also used in Overeem et al. (2016) for sensitivity analyses. The reviewer is right that this dataset is likely more rainy than an arbitrary period of 12 days, but this does not imply that it rains all the time. We expect that this 12-day period contains more dry than rainy time intervals. The exact rate of wet-dry 15-min intervals for the calibration and validation datasets has been derived from the path-averaged radar data and is mentioned in our revised manuscript.

We wrote:

L274: "...Although the calibration dataset has been selected considering rainy days, the number of non-rainy data points is much higher than the number of rainy data points, representing 93%, which is comparable to the average occurrence of dry spells in the Netherlands according to automatic weather stations..."

L353: "...According to the wet-dry observations during the validation period, we observed that 97% of the data points represent non-rainy intervals. Being just percentage points higher than for the calibration period, the periods can be considered comparable to each other. Moreover, the employment of the MCC metric justifies any wet-dry distribution dissimilarity...".

**2. Usage of questionable classification metric, Simple Matching (SM):**

**The Simple Matching (SM) is chosen as performance metric for the binary classification into wet and dry periods. SM, which is the same as Accuracy (a more common term for this metric for binary classification performance) is very sensitive to the balance of positive and negative samples, see my comment on L187 for an explanation. In general Accuracy is thus not a recommended, but still widespread, metric. More info can be found e.g. here https://dx.doi.org/10.1186%2Fs12864-019-6413-7. This article recommends to use the Matthews correlation coefficient (MCC), which I would also recommend. Other options would be to study the ROC curve, or to be more careful with balancing wet and dry samples in the calibration and validation period. I strongly recommend to redo the optimisation of the wet-dry parameters, taking all this into account.**

Reply:

We admit that this issue, which is one of the critical issues that should be discussed, was not addressed. We now optimize by employing MCC as performance metric. In spite of being a conceptual metric for imbalanced classification problems, the use of MCC in the wet-dry classification process did not resulted in improvements for the rainfall retrieval process, significantly. MCC is used indirectly in rainfall retrieval, where the best parameters obtained by maximizing MCC value in wet-dry classification process are used to calibrate rainfall retrieval process parameters.

Figure 5 and Table 5 summarize the RAINLINK performance.

**3. Unclear method for determining optimal wet-dry parameters:**

**There is another problem with the optimisation of the wet-dry parameters. The optimal parameters are not those that clearly provided the highest values of SM, see my comment on L225 and on Fig 2a. It is not 100% clear to me how the optimal parameters are derived. If they are derived from the "behavioral" solutions I find this problematic, because these distrubtions are somewhat arbitrarily selected, see my comment on L225 for a more detailed explanation. I might, however, not have fully understood how the optimal parameters are found. In this case, please explain the method better and also, in particular, explain why not the parameters at the best SM values are chosen. Of course, as**

**stated above, SM is not a good metric for judging wet-dry performance. Hence, in case a different metric is used, things will look differently here anyway.**

Reply:

Thanks for the comment and recommendation. We run again the wet-dry calibration by using the recommended performance metric (MCC). Now we derived the values for the optimum solutions and presented the median values of the parameters considering the distribution of the "behavioral solutions". The idea is to show the uncertainty associated with the best parameters set, once there are many optimal solutions.

We wrote:

L253: "…The parameters $WD_{p1}$ , $WD_{p2}$ , $WD_{p3}$ ,$WD_{p4}$ , and $WD_{p5}$ reach the optimum values equal to 7.5 h, 14.1 h, 19.7 km, −2.7 dB, and −0.9 db km$^{-1}$ , respectively…"

Also, Fig.2 was updated and now seems to find a reasonable optimal region.

**4. Missing validation of wet-dry classification:**

**The validation section is completely missing a validation of the wet-dry classification. Given the issues with identifying the parameters of the wet-dry classification and its potential impact on rain rate estimation (see my comment on L311), it is crucial to add it here, also including an analysis of its impact on rainfall sums.**

Reply:

We agree with this observation, a validation of the wet-dry classification during the validation period would complete the manuscript. Hence, we added a section dedicated to wet-dry classification for the validation period, which shows that the use of MCC instead of SM gives slightly better results:

L344: "…3.2.1 Wet-dry classification validation…"

**5. Unclear motivation of the proposed calibration:**

**It should be made clearer why the calibrations that have been done in other RAINLINK publications are not sufficient. Furthermore it should be made clearer why LH-OAT and SPSO have been selected, highlighting and explaining their advantage compared to past calibration efforts. (See my specific comment on L79)**

Reply:

We appreciate this observation and we explained the added value of our approach compared to those in other RAINLINK publications as well as why LH-OAT and SPSO have been selected. The main idea is to highlight how a stochastic and pinpointed calibration approach can be more parsimonious, reducing computational demand and driving the algorithms to a better performance. An advantage is that all RAINLINK parameters are initially taken into account, whereas previous studies focus on a limited set of parameters. Moreover, the optimization of the wet-dry classification is separated from the rainfall retrieval, i.e., first the wet-dry classification is optimized, next the rainfall retrieval.

We wrote:

L85:"...In fact, many optimum solutions can occur, in accordance with a strong variability of the parameters, thus the optimization should account for the distribution of these solutions and parameters, selecting them based on uncertainty levels..."

L94: "...also we optimize for the first time the main RAINLINK processes, namely wet-dry classification and rainfall retrieval, separately..."

L170: "... Having the same feature as the Monte Carlo sampling, i.e., a global screening method, LH sampling reduces the computational cost significantly (n -1 times), being more efficient (Van Griensven et al., 2006)..."

L179: "...After having selected the most important parameters by sensitivity analysis, the RAINLINK parameters are optimized with the method Standard Particle Swarm Optimization (SPSO-2011) (Clerc, 2012). Being a major improvement over previous

PSO versions, with an adaptive random topology and rotational invariance, SPSO-2011 is a stochastic, effective, and efficient calibration method, as highlighted in recent studies with other hydrological and environmental models (Abdelaziz and Zambrano-Bigiarini, 2014; Bisselink et al., 2016; Pijl et al., 2018)..."

**Additional note:**

**In the light of the (according to my interpretation of the presented results) large impact of false-positives on PBIAS, one could (or maybe should), consequently calibrate the rainfall estimation part of the algorithm with taking the wet-dry classification from the reference to avoid an overestimation of wet antenna attenuation that has to compensate the long-term rainfall overestimation from false-positives. This is just an idea that, assuming that large parts of the analysis have to be redone for a revision, could be explored.**

Reply:

Good point, we will consider this possibility for a future study.

We wrote:

L463: "...Moreover, due to the large impact of false positives on PBIAS, a calibration of the rainfall retrieval process taking into account the wet-dry classification from the reference should be considered for further research. Thus, an overestimation of wet antenna attenuation that has to compensate for the long-term rainfall overestimation from false positives would be avoided..."

**Specific comments:**

**L22: One has to be careful with the interpretation of the number of stations available in GPCC. Large delays in data delivery and data processing lead to a delayed peak of available stations. From how I interpret the GPCC documentation, this might explain most of the "decline" since the 1980. The GPCC authors write "The decrease of the number of stations from more than 45,000 in 1961-2000 down to 10,000 stations after 2019 is caused by the delay of the data delivery to and by post-processing at GPCC" (Source: https://opendata.dwd.de/climate_environment/GPCC/PDF/GPCC_intro_products_**

**v2020.pdf, end of page 9). Hence, this sentence should be reformulated accordingly.**

Reply:

Thanks for the observation, we corrected this as follows:

L21: "...the Global Precipitation Climatology Centre (GPCC), underwent a decline caused by the delay of the data delivery and by post-processing at GPCC. A reduction of approximately 43,000 (81%) and 27,000 (77%) rain gauges with monthly and daily precipitation records during the last 30 years, respectively (Schneider et al., 2021)...."

**L48: Providing the information about the study area for Chwala et al. (2012) is a bit misleading here, because they did not study spatial rainfall information. Hence, the very low CML density in this study that is listed here, was not a relevant factor.**

Reply:

Thanks for your suggestion, we removed the reference to Chwala et al. (2012) in the text.

**L60: Since pycomlink contains different algorithms, of which Graf et al (2020) only used a selection, I would write "...rainfall retrieval packages" here.**

Reply:

We modified this accordingly.

We wrote:

L57: "...Long-term studies involving country-wide verification of CML rainfall estimates based on data from a few thousand CMLs are provided by Overeem et al. (2016b) for the Netherlands employing RAINLINK (Overeem et al., 2016a), and by Graf et al. (2020) for Germany employing pycomlink (https://github.com/pycomlink/pycomlink), both open-source rainfall retrieval algorithms packages..."

**L79-L81: I do not understand the argumentation here. If one can get the "most precise path-averaged rainfall intensity estimates" using the optimised parameters from the empirical calibration, why is a new calibration needed. Aren't the old RAINLINK calibration enough? Maybe this should be improved together with the parts around L87. It is not clear what the drawbacks of the "deterministic" calibration of RAINLINK are. Since this is the core motivation of this work, I recommend to make this clearer here.**

Reply:

What we would like to highlight is that we have many similar "best" solutions for the optimized parameters (sometimes referred to as equifinality (Zambrano-Bigiarini et al., 2013)). Employing a stochastic approach allows us to access the uncertainties associated with the best set of parameters. See also our reply to comment #5.

**L104: How have the 12-days been selected in this period from June till September 2011? In case you only select rainy days, you skew the average distribution of wet and dry data points. This shifts your false-positive and false-negative rates in the validation period compared to the calibration period. Hence, the optimal wet-dry parameters from the calibration period might not be optimal for the validation period.**

Reply:

We selected summer rainy days. However, the fractions of dry periods are relatively similar to each other: 93% and 97% non-rainy data points are observed for the calibration and the validation period, respectively. Also, we used a different performance metric, MCC, to reduce the distribution mismatch problem. See our reply to comment #1.

**L155: It would be nice to learn a bit about the computational demand of the sensitivity analysis.**

Reply:

We thank you for your interest and added more information about the computational advantage of the LH-OAT method.

We wrote:

L170: "…Having the same feature as the Monte Carlo sampling, i.e., a global screening method, LH sampling reduces the computational cost significantly (n -1 times), being more efficient  (Van Griensven et al., 2006)…"

**L165: How is this relative importance related to the parameter range that was selected. Without understanding the details of the LH-OAT method, I can imagine that the parameter range influences the step size and hence the relative impact of each step. Please comment (or just correct my wrong assumptions on how LH-OAT works…).**

Reply:

The step size is selected as a fraction of the parameter range. In this manuscript the fraction was set to 0.1, which is the default of the R package hydroPSO. We added this information in the text.

We wrote:

L176: "…We choose a step size that represents a fraction of 0.1 of the parameter search space…"

**L169: Why was SPSO selected? What are the advantages, also compared to other optimisation methods? What are potential disadvantages?**

Reply:

The Standard Particle Swarm 2011 (SPSO2011) used in the manuscript is a recent member of the calibration/optimization family in water resources, which is more efficient than DREAM, SCE-UA and other well-known algorithms, as observed in Zambrano-Bigiarini et al. (2013).

SPSO-2011 is a major improvement over previous PSO versions, with an adaptive random topology and rotational invariance constituting the main advancements.

We wrote:

L180: "...Being a major improvement over previous PSO versions, with an adaptive random topology and rotational invariance, SPSO-2011 is a stochastic, effective, and efficient calibration method, as highlighted in recent studies with other hydrological and environmental models (Abdelaziz and Zambrano-Bigiarini, 2014; Bisselink et al., 2016; Pijl et al., 2018)..."

**L178: Why was simple matching chosen as metric for the binary classification? It seems to be sensitive to unbalanced distributions of true and false values. E.g., if, in the case of wet-dry classification, the number of dry data points is by far larger than the number of wet data points, very high values of SM can just be reached by setting everything to "dry".**

Reply:

We thank you for your constructive view and we redid the analyses by considering a proper performance metric (MCC). We redid all the analyses considering the use of MCC. See our reply to comment #2.

**L178: Here it sounds as if the modified KGE is used as metric for the wet-dry classification. This should be rephrased.**

Reply:

We split the algorithm processing into wet-dry classification and rainfall retrieval itself. Thus, we employed the KGE metric only for the rainfall retrieval process. We made this clear in our revised manuscript.

We wrote:

L189: "…The goodness-of-fit measures chosen to drive the optimization and performance for the wet-dry classification and the rainfall retrieval processes are the Matthews Correlation Coefficient (MCC) (Matthews, 1975) and the modified Kling-Gupta Efficiency (KGE) (Kling et al., 2012), respectively…"

**L180: I guess the gauge-adjusted radar product comes with 0.01 mm resolution or similar. The path-averaging along the CLM paths results in even smaller values. Wouldn't it makes sense to define a threshold slightly above zero to divide between wet and dry periods because something below 0.1 mm in 15-minutes can hardly be considered rain?**

Reply:

Yes, we used a threshold of 0.25 mm to classify intervals as rainy when above this value.

We wrote:

L192: "…A 15-minute time interval from a given sub-link is considered dry if the reference is below 0.25 mm… "

**L182: "…where d is the number of links classified correctly as dry…". I expected that this is done for all data points and not for each link. If this is done for each link, that would mean SM is calculated for each time step. But in the context of this work, it seems to be calculated for all samples for the whole calibration period, correct? Please clarify.**

Reply:

Thanks for giving attention to this aspect. Yes, this was calculated for all data points. We rephrased this part, as well as the change to the MCC metric.

We wrote:

L194: "...Due to the higher frequency of non-rainy 15-min intervals (data points), the process of wet-dry classification is considered an imbalanced classification problem. Employing recurrent metrics for binary classification, such as F1 score and accuracy, may lead to inflated results. The Matthews Correlation Coefficient is less subjective and preferred since it informs how correlated the predictions and observations are, reaching a high score only if the prediction obtained good results in all the four confusion matrix categories (true positives ($TP$), false negatives ($FN$), true negatives ($TN$), and false positives ($FP$)) (Chicco and Jurman, 2020). The Matthews Correlation Coefficient is defined as..."

**L203: It is not clear here if the "mean rainfall over the Netherlands" is based on interpolated rainfall maps, or the average of the rainfall values for each CML.**

Reply:

Here we mean the average of the rainfall values for all CMLs, and we made this clear in our revised manuscript.

We wrote:

L232: "...daily mean rainfall over the Netherlands estimated from the CML values (as time series..."

**L220: What is "behavioral" supposed to mean here?**

Reply:

Here "Behavioral" is the set of solutions which presents the best performances. We removed this term in this manuscript part and defined it after (Zambrano-Bigiarini et al., 2013).

**L225: I do not understand how the optimal values have been identified. The only metric that is used here is SM. Hence, I expected to find the optimum where the cyan coloured dots (highest SM) in Fig 2a are. The parameters reported in the**

**text are, however, more in the centre of the parameter range, while the highest SM values are at the smallest WD_p4 and highest WD_p1 values. Maybe this has to do with the "Wilcoxon signed rank test" that is mentioned in the sentence before. I could imagine that the derivation of the optimum is somehow based on the distribution of "behavioral" solutions. But, since the distribution of "behavioral" solution heavily depends on the arbitrarily chosen threshold of SM, this is not a reliable procedure. If the SM threshold would be set to e.g. 0.95, the distribution would look very different and for WD_p1 show a clear tendency towards very high values. In conclusion, I find the results very counterintuitive. Please either provide a good explanation for the chosen method or correct your procedure of determining the optimum. Please note that using SM is not a good choice anyway, see my comment on L178. Hence, potentially redoing this step of the calibration should then be done with a different performance metric.**

Reply:

Thanks for pointing this out. Now, we considered the optimum value and identify it accordingly. Moreover, we consider the median values for the "behavioral solutions", being selected for the class of solutions for which the performance was greater than 0.53. The median values were considered to highlight the variability associated with the likely solutions and the uncertainty associated with the "best" choice. Please see our reply to comment #3.

**L229: What is the point of the 95% confidence interval of the "behavioral" solutions? Or maybe more general, what is the point of the "behavioral" solutions, which have been obtained by arbitrarily selecting solutions with SM larger than 0.90? Why not use SM > 0.95 as threshold?**

Reply:

We appreciate your observation. We decided to remove the confidence intervals from the manuscript. We now present the optimum and the median of the best solutions.

**L232: If I understand the analysis correctly, a SM of 0.9 for the whole calibration does not mean that "90% of the microwave links provide a correct wet-dry**

**classification considering the entire period of 12 days". I would rather say that 90% of the data points are classified correctly. It is not clear how these correct classifications are distributed between the individual CMLs. Maybe I do not understand how SM is calculated here for the calibration periods (see also my comment on L182). Please clarify.**

Reply:

Perfect point. You are right this is related to the data points and not to the CMLs themselves. We changed the performance metric from SM to MCC and removed this statement from the manuscript.

We wrote:

L263: "…The values obtained for the calibrated parameters are based on the median of the "behavioral" solutions and are in line with the default parameters, except for $WD_{p2,}$ which indicates a smaller period for computing the maximum of the minimum received power…."

**Fig 2a: I find it strange that very high SM values are more or less equally distributed over the full range of WD_p5, but WD_p5 is considered the parameter with the highest relative importance according to Table 3. How can that be explained?**

Reply:

As observed previously, this was related to the performance metric characteristic. We redid the analyses by using the MCC metric and now the $WD_{p5}$ distribution seems to find a clear optimum (Fig. 2). Please see our reply to comment #3.

**L260: If the optimisation is done only with rainfall data at the CMLs and not on CML-derived rainfall maps, I do not see how an optimisation of the outlier filter can be done. Assuming that there are a few outstandingly good performing**

**CMLs, all others would be removed in the process, because this would results in the highest average KGE. Please make this clearer in the text.**

Reply:

Thanks for giving attention to this aspect. We made this clearer and added a brief discussion. A sensitivity analysis for the outlier filter can be carried out (Overeem et al, 2016), but it is indeed difficult to take the related threshold parameter into account in the optimization.

We wrote:

L298: "...One way forward to calibrate $RR_{p3}(F_t)$ would be to include both the number of available links in the optimization or perform an optimization based on rainfall maps, which can be influenced by the underlying CML network density..."

**L269: "...which is in line with what can be seen in Fig. 3.". I find it interesting that this is the case here but not for Fig. 2a. Please explain (which is maybe already done in response to my comment on L225).**

Reply:

Exactly, the results in Fig. 2a. were counter-intuitive. However, we redid the analyses and now the graphics are more intuitive.

**L270 and following: I find it most striking that there seems to be a clear correlation between RR_p4 (wet antenna attenuation) and RR_p5 (alpha). The explanation probably is that a higher alpha leads to higher rain rates, because the weight of the maximum attenuation increases, which has to be compensated by a higher value of wet antenna attenuation correction, decreasing the rain rate estimates. Hence, this two parameters clearly influence each other. This should be mentioned in this section.**

Reply:

We appreciate your recommendation and added this likely relation between the parameters.

We wrote:

L318: "...These parameters are expected to be positively correlated. Likely, higher $RR_{p5}$ values lead to higher rain intensities, increasing the weight of the maximum attenuation and consequently a higher value of $RR_{p5}$ would become necessary to compensate for the extra attenuation, decreasing the rain intensity estimates..."

**L285: The validation of the wet-dry classification seems to be missing completely here. I strongly suggest to include it, in particular because I expect the results to be very different from the calibration period because of the different ratio between wet and dry data points in the two periods and because SM is not robust to changes in this ratio.**

Reply:

Yes, we added the wet-dry classification for the validation period. See our reply to comment #4.

**L303: It would be nice to see a figure similar to Fig. 4 also for the 15-minute data. I am aware that similar plots have been shown in several RAINLINK publications, but, it would be interesting to see the differences between default and calibrated processing not only for the daily data.**

Reply:

We appreciate your suggestion. We already present the metrics for the 15-minute data in Table 5. We understand that a scatter plot could present the dispersion around a reference line. However, as we study different assessment thresholds in Table 5, we believe that a figure with similar results could oversize the manuscript. Hence, we decided to not include such scatter plots and we hope the reviewer understands.

**L304: "For a complete evaluation we use different rainfall thresholds." It took me some time to understand this sentence. If the reader does not already know the details of Table 5, it is not clear what the "complete evaluation" is and what the "different rainfall thresholds" are used for.**

Reply:

Well observed, we rephrased this part.

We wrote:

L368: "…Next, the performance of 15-minute path-averaged rainfall estimates is investigated. Table 5 summarizes RAINLINK's performance when the default and calibrated parameters are applied for different rainfall thresholds…"

**L311: My explanation for the strong influence of the threshold "Reference > 0" on PBIAS is the following. There is most likely a large number of false-positives. These false-positives contribute significantly to the overall CML-rainfall estimates and result in a positive PBIAS. This impact of false-positives on the CML rainfall estimation is nicely shown in Fig 9. in Polz et al. (2020, https://doi.org/10.5194/amt-13-3835-2020). If the false-positives are removed, which is what the threshold "Reference > 0" does, the resulting CML-rainfall estimates are missing this large amount of "false-positive" rainfall. As a consequences, PBIAS shows a strong underestimation of CML rainfall estimates. This effect also explains the other observations, made in the sentences before. The fact that PIBAS is is "better" for the calibrated parameters turns into a disadvantage when applying "Reference > 0", because the shift of PBIAS towards underestimation seems to be similar for calibrated and default parameters (explaining the observations in L307). The reason why the effect on PBIAS does not appear when applying a threshold like "Reference OR RAINLINK > 0" is that this threshold does not remove the false-positives, because if RAINLINK > 0 and Reference = 0, the data point is kept in the dataset. Your sentence in L309 "This underestimation is not observed if both RAINLINK and the reference are above the threshold" is not correct, because you apply an OR not an AND for these**

**threshold. As stated above, I strongly recommend to include an analysis of the wet-dry classification for the validation data. Furthermore, as stated in my comments on the calibration of the wet-dry classification, the choice of parameters might not be optimal for the calibration period. Hence, there might also be less impact of false-positives, if another "optimal" parameter set is found.**

Reply:

Thanks for the constructive observation. We added a wet-dry classification validation section and employed the MCC metric. We rewrote the paragraph taking into account the appropriate interpretation.

We wrote:

L371: "…As for PBIAS, the default parameters outperform the calibrated ones for the thresholds "Reference > 0" and "Reference > 1", whereas the calibrated parameters show better performance for the remaining thresholds. One can also observe that, if a threshold is only applied to the reference and consequently the false positives are removed, RAINLINK shows a large underestimation with respect to the reference. This underestimation is not observed if either RAINLINK or the reference are above the threshold…"

**L317: I guess you are referring to Table A1 in de Vos et al. (2019). There seems to be a typo, either in this table or in the sentence here, because in the Table A1 the Pearson correlation for the revaluation is 0.27 and not 0.52 as written here.**

Reply:

Well observed, it is a typo, which we corrected.

**L320: Since the reevaluation covers winter months and since this is know to introduce overestimation of CML rainfall estimates, I would have guessed that de Vos et al (2019) have a high bias in their analysis, which apparently is not the case. Please explain a bit more detailed where this difference in PBIAS could**

**stem from, because I do not understand how "different periods, with different durations" lead to the high PBIAS in this study compared to de Vos et al (2019).**

Reply:

Note the we refer to the 613-day evaluation of the min/max sampling in Table A1 from de Vos et al (2019), which only contains data from 18 Feb - 16 Oct, so the influence of winter months is limited. The data are from 2011, 2012 and 2013 and also include the 3-month period used in this study. We find it difficult to explain the differences in PBIAS. We added a possible explanation to the manuscript:

We wrote:

L389: "…Possibly, the wet-dry classification using default parameters applied by de Vos et al. (2019) results in less false positives or due to the longer period the false negatives compensate for the false positives, resulting in a lower PBIAS value. The summer of 2012 was rainy, with 286 mm of rain compared to the climatological average of 225 mm, averaged over the Netherlands. For the central weather station in the Netherlands, a long precipitation duration of 153 hours was observed compared to the climatological average of 121 hours over the summer months June, July & August. This could be a reason for differences in PBIAS, although this summer is also part of the 613-day dataset evaluated in de Vos et al. (2019)…"

**L327: As explained in my comment on L311, I assume that false-positives play an important role for the overestim ation of CML rainfall.**

Reply:

We agree and wrote

L400: "…This overestimation observed in the double-mass curves is in line with the PBIAS values reported earlier (Tab. 5), being justified by the higher presence of false positive observations…"

**L334: Why is this not done with rainfall maps, which are also easily produced with RAINLINK? That would be a more relevant basis for doing an analysis "over the Netherlands".**

Reply:

Thanks for giving attention to this aspect. Actually, we intend to evaluate the path-averaged rainfall when distributed over an area and the associated error behavior. Evaluating a spatially interpolated map, the error of interpolation process would be added in the analyses, which is beyond the scope of this work.

**L337: I do not understand how the area plays a role here. You average the data from the individual CMLs, not taking into account how they are distributed over this area. The effect on PBIAS and beta has nothing to do with the fact that the CMLs are within a certain area.**

Reply:

We rephrased this text as follows

L411: "...Since the CML rainfall estimates are averaged over a ~35,500 km$^2$ area not taking into account how they are distributed, the PBIAS and β values stay the same (Fig. 5)..."

**L339: I would not call this an "areal time series". On could maybe argue that an sensor-average from a fairly homogeneously distributed rain gauges network is representative of certain area, but not an average of a very heterogeneous sensor network like the one of the CMLs here.**

Reply:

Interesting observation. The CML coverage over the Netherlands is not homogeneous indeed and those urban areas with high network density will have a larger weight in the computation of the areal rainfall. On the other hand, the number of CMLs is much higher than those from official rain gauge networks, i.e. they could

provide a better spatial average than gauge data. Hence, we think that calling this "areal time series" is justified, although we added the above-mentioned limitation to our revised manuscript. Studying the calibration for different classes of CML features would be an excellent topic for a future researcher.

We wrote:

L414: "…In spite of not being a homogeneous network, the CMLs are observing in the entire Netherlands, having a high enough spatial representativity for computing a spatial average rainfall…"

**L347: Shouldn't one reason for the differences between calibrated and default parameters be that the calibration here is done with a more sophisticated, presumably better, method?**

Reply:

Yes, this is likely one of the reasons and we added this to our revised manuscript.

We wrote:

L425: "…Moreover, the calibration here is done with a state-of-the-art and efficient method…"

**L349: I would add WD_p1 and WD_p4 here, because Fig 2a shows that the highest values for SM are reached at the end of their parameter range. Hence, it can be expected that SM could further increase beyond the current parameter range if it would be extended. So the questions is, why was this not done.**

Reply:

We redid the wet-dry classification using a MCC metric and extended the parameter range for $WD_{p1}$ and $WD_{p4}$.

**L372: I can not follow this argumentation. While I agree that "hydrological and meteorological scales of application are defined over areas", I would say that these scales, in particular in hydrology, are much smaller than the Netherlands for which the positive effect of aggregation over an area is found in this manuscript.**

Reply:

The reviewer is right that hydrological scales in the Netherlands are much smaller than the country. However, we would like to highlight the CMLs' power to monitoring rainfall over areal scales in a general manner. Thus, we rephrased this as:

L449: "…This result is important, because from a general perspective, hydrological and meteorological scales of application are defined over areas, e.g., watersheds, climate zones, political and administrative regions, etc.…"

**L389: Just a comment. Yes, comparing to gauges avoids the impact of radar errors, but the path-averaged nature of CMLs has to be considered when comparing to rainfall data from point observations. Furthermore, since the gauges would have to be fairly close (maybe less than 2km) to be able to assure comparability with CMLs on 15-minute or 1h time scales, this would limit the number of CMLs that can be analysed.**

Reply:

Yes, excellent point, a perfect reference to evaluate CML rainfall estimates is a challenge yet. Also note that representativeness errors in radar data are a limitation when providing path-average reference data.

**Editorial comments:**

**L131: Maybe write "summarises" instead of "highlights" here.**

Reply: We replaced "highlights" by "summarises".

Final remarks:

We thank the reviewer for the kind comments about the contribution to the field and the paper being well organized and interesting to read. Basically, we accepted all the reviewer's contributions (except one) and hope that we reformulated the manuscript following the same quality, care and effort employed by the anonymous reviewer.

References

Overeem, A., Leijnse, H., and Uijlenhoet, R.: Retrieval algorithm for rainfall mapping from microwave links in a cellular communication network, Atmos. Meas. Tech., 9, 2425–2444, https://doi.org/10.5194/amt-9-2425-2016, 2016.

Zambrano-Bigiarini, Mauricio, and Rodrigo Rojas. "A model-independent Particle Swarm Optimisation software for model calibration."*Environmental Modelling & Software* 43 (2013): 5-25, https://doi.org/10.1016/j.envsoft.2013.01.004.

---

## Author Response (AR2)

**Author's replies to the referee's comments to manuscript AMT-2021-34**

**Please see the file with track changes to check the edited lines.**

The original referee's comments are written in bold and the author's replies are written in regular font.

**Anonymous Referee #1, revision 2**

**The authors addressed the two main concerns I have previously raised. Indeed, I would still like to see a more thorough discussion referencing the relevance of the power-law coefficient calibration as a complement method with respect to the presented approach (e.g., there have been studies presented different ways to calibrate the power-law coefficients for a specific climate zone using the CML's standard min/max measurements and a rain-gauge, and not only dis-drometer-based calibration). However, since this issue is now somewhat mentioned in the paper, it is not a major issue anymore.**

Dear Referee #1 , we appreciate our revision and wrote the following sentence related to power law coefficients.

329-335: "…A physically-based approach which derives these coefficients from drop size distribution observations and scattering computations is preferred compared to optimizing these coefficients in a statistical manner. Especially, for frequencies higher than 35 GHz. The drop size distribution dependence of the k-R relation in the frequency range of approximately 20-35 GHz is considered small compared to errors from wet antenna attenuation or erroneous wet-dry classification. Although a physically-based approach is considered better, a calibration of power-law coefficients may be a way forward for regions which lack disdrometer data (Ostrometzky and Messer, 2020)…"

**Anonymous Referee #2, revision 2**

**The authors have provided a substantial major revision. They have redone their analysis according to my suggestions and have updated the manuscript. The updated analysis did not change the results much, but the methodology is now much more sound. My only complain is that the newly added text is often not of good quality and hence should be revised carefully. In summary, I only have some minor and specific comments that can be addressed in a minor revision.**

We gratefully thank the Referee for the constructive comments and recommendations. We carefully checked the quality of the added text.

**Minor comments:**

**1. The impact of false-positives and false-negatives that results from the wet-dry classification could be discussed in more detail. Besides providing optimized RAINLINK parameters, this manuscript also shows the sensitivity of the wet-dry classification to its parameters. Most important, it also shows the challenge of wet-dry classification and its limitations, i.e. there is still a significant number of false positives and false negatives even after optimisation. This is important information and hence should be pointed out more clearly. See also my comment on section 3.2.1.**

Reply:

We rewrote this section giving attention to the limitations and the impact of false-positives on rainfall estimates. Please see your comment on section 3.2.1.

**2. I am missing a table with the optimised parameters so that they are quickly to grasp. Maybe this info could just be added to table 1 and table 2. Or the optimised parameters could be shown together with their performance metrics in comparison to the default values and those from other RAINLINK calibrations. The later option might be hard to put into one table without making it confusing, though.**

Reply:

We added a Table in the conclusions section. Now the values of parameters can be checked easily.

**Specific comments (line numbers refer to the diff version):**

**L19-23: This still sounds as if there is an ongoing decline of rain gauges that is evident from the data availability plot of GPCC. Writing that the GPCC database "underwent a decline" sounds as if GPCC would get less and less data. From how I understand the last GPCC report that I referenced in my last review, this is not true. There is a constant increase of data. The largest portion arrives at irregular intervals and with large delay, though. There might be a global decrease of rain gauges which are in operation, but**
**the GPCC data availability plot cannot be used to deduce such a trend. I suggest to reformulate this section once more.**

Reply:

We reformulated this section given attention to the constant increase of the GPCC database. We added the following sentences:

L19-26: "... Another issue is the data availability of ground-based measurements. For instance, the largest worldwide rain gauge database, maintained by the Global Precipitation Climatology Centre (GPCC) had 45,000 rain gauges in 1961-2000 and down to 10,000 after 2016. This decrease was caused a delay in data delivery and by post-processing at GPCC (Schneider et al., 2021). Although, decreasing in the past due to quality control, the GPCC database has been increasing in recent years as a result of delivery of updates as well as supplements with additional stations and long time-series of data (Schneider et al., 2021)..."

**L62: I would not say that "data-driven solutions are not feasible for places or countries without sufficient reference data". The training can, of course, only be carried out in regions with sufficient reference data. But the trained methods, like the ones in the references that you cite, can potentially be used with data from any region. Transferability can be questioned, though. But this is also true for most other CML processing methods which are typically developed with data from only one climatological region. The big disadvantage I see with data-driven solution is that you cannot readjust them to a new dataset just be tuning two or three parameters. I suggest to slightly rephrase this new section.**

Reply:

We rewrote:

L63-69: "...These data-driven solutions also hold a promise for ungauged areas, but it will not be feasible for places or countries without sufficient reference data to train the machine learning algorithms. That is, data-driven models require a huge number of observations to learn and detect the whole behavior of the phenomenon to be modeled. For other algorithms, such as RAINLINK, it may still be feasible to at least tune a few parameters, for instance, by employing drop size distribution observations (from a region with a similar climate) to obtain more appropriate coefficients of the relationship between specific attenuation and rain rate..."

**L85: This new sentence is hard to understand. Please reformulate and/or split into two sentences.**

Reply:

We rewrote and tried to simplify the sentence as:

L89-91: "...In fact, many optimum solutions can occur, corresponding to different parameters sets (a phenomenon known as equifinality)..."

**L94: the part "..., also we..." does not sound like correct English to me. Anyway, a new sentence could be started here.**

Reply:

We modified this as follows:

L100-101: "...Moreover, we optimize for the first time the main RAINLINK processes, i.e., wet-dry classification and rainfall retrieval, separately..."

**L176: My question from the last review is still not answered: "How is this relative importance related to the parameter range that was selected?" Let's say, you select a too small parameter range because you do not yet know the sensitivity. Then the "relative partial effect", which as far as I understand, will depend on the absolute step size, which will be very small for the too small parameter range. So my question is not, what is the relative step size, but how the parameter range, which influences the absolute steps size, could impact the importance of a parameter in this analysis.**

Reply:

This is an important point, however, we have not tested different parameter ranges to obtain such insight. We delimited the ranges based on expert judgment and trial runs and believe that the employed ranges are quite wide. We can consider this question as a research gap to be analyzed in the future.

We wrote in the conclusion section:

L492-495: "...Further research can be conducted to test how the parameter range affects the importance of parameters in this approach. Specifically, even wider parameter ranges could be tested. Moreover, a longer calibration period could be analyzed to make the optimized parameters more generally applicable to other data from other periods. This especially holds for the wet-dry classification process.

**L241-244: Since WD_p2 is now by far the most important parameter, this should be explained in the text. I would also like to understand why WD_p2 suddenly is so much more important than before.**

Reply:

Well observed. Actually, it surprises us a bit that WD_p2 is now the most important parameter in the sensitivity analysis. We find it difficult to explain this. We wrote:

L244-247: "...The highest importance reached by the $WD_{p2}$ parameter highlights the rain-induced attenuation temporal correlation. Since, this parameter represents the number of previous hours over which the maximum value of the minimum received power ($P_{min}$) is computed, it governs the wet-dry classification process by influencing on the attenuation (median($\Delta P$)) and specific attenuation (median($\Delta P_L$)) computation..."

**L249: Remove "the" before "all solutions"**

Reply:

We wrote:

L253: "The distributions are obtained for all solutions..."

**Fig 2.: It is hard to see relations between the different WD parameters. I suggest to rearrange the individual plots to a scatter plot matrix, e.g. using https://ggob-i.github.io/ggally/reference/ggpairs.html, because this way all relations and potential correlations would be visible. The distributions that are now shown on the bottom, would also fit on the diagonal in the scatter plot matrix.**

Reply:

Excellent point. We rearranged this Figure by using the ggpairs function.

**L271: "Due to the similar value of WDp1..." I do not understand this sentence. Why are data excluded due to similarity of WDp1 values?**

Reply:

We wrote:

L269-273: "...This parameter has a direct relation with data availability, since it determines the minimum number of hours needed to compute max($P_{min}$). Note that max(Pmin) is only computed if at least a minimum number of hours of data are available; otherwise it is not computed and no rainfall intensities will be retrieved (Overeem et al., 2016b)."

**Table 4: Similar to table 3, what is the reason that the order of relative importance changed and that there is clear leader, RR_p5 here?**

Reply:

We wrote:

Well observed, we identify that all ranks changed in our analyses. We presume that the change of the cost function (MCC now) is the reason for that. Although MCC does not increase the calibration performance, this metric seems to be a better choice, because of the imbalanced feature observed in this dataset. A possible explanation is that the overall effect of the attenuation correction, 1.74 dB, is rather small compared to the attenuation due to rain. Hence, the actual derivation of mean rainfall intensity from minimum and maximum rainfall intensity may be dominant.

**L309: I do not understand what "bears to similarity" means.**

Reply:

We wrote:

L306-307: "...the optimum values, 1.7 and 0.23 are almost identical with the median values for the "behavioral" solutions, 1.74 and 0.24..."

**L320: Here you probably mean RR_p4 and not RR_p5.**

Reply:

Yes, you are right and we modified this.

**L330-336: It could be noted here that the k-R relation is not very sensitive to DSD variations for frequencies in the range of approx. 20-35 GHz. Compared to errors from wet antenna or wrong wet-dry classification, the DSD dependence of the k-R relation in this frequency range can be considered to be small.**

Reply:

We wrote:

L329-335: "...A physically-based approach which derives these coefficients from drop size distribution observations and scattering computations is preferred compared to optimizing these coefficients in a statistical manner. Especially, for frequencies higher than 35 GHz. The drop size distribution dependence of the k-R relation in the frequency range of approximately 20-35 GHz is considered small compared to errors from wet antenna attenuation or erroneous wet-dry classification. Although a physically-based approach is considered better, a calibration of power-law coefficients may be a way forward for regions which lack disdrometer data (Ostrometzky and Messer, 2020)..."

**L346: The MCC of 0.4 for the validation is significantly smaller than the minimum MCC of 0.53 of all "behavioral" solutions from which the mean parameters where taken. What is an explanation for this strong decrease in performance?**

Reply:

We wrote:

L347-353: "...We find a MCC value of 0.4 for the validation dataset, being smaller than the MCC threshold for "behavioral" solutions, i.e., 0.53. This occurred because the calibration did not generalize at all the wet-dry classification process. It was focused on the calibration dataset, capturing many details and noise, and subsequently failed to capture a different trend from another dataset, i.e., became an overfitted model. Thus, the performance for the validation dataset was worse, because the calibration dataset will not be entirely representative for other periods. A solution could be to increase the size of the calibration dataset, encompassing more characteristics and trends about the phenomenon..."

**L353: Maybe write "wet-dry observations of the reference" to make it clear that these labels are derived from the reference.**

Reply:

We wrote:

L361: "...According to the wet-dry observations of the reference during the validation period..."

**L353-355: While 97% (number of dry data points in validation period) and 93% (number of dry periods in calibration period) are numbers which seem close to each other, I want to point out that the relative number of wet periods is more than twice as high in the calibration period (7%) compared to the validation period (%3). I am, however, not sure about the exact impact on the results, e.g. the significantly decreased MCC in the validation periods. You might want to think about this issue and add a comment to the text.**

Reply:

We wrote:

L361-366: "...According to the wet-dry observations of the reference during the validation period, we observed that 97% of the data points represent non-rainy intervals. Being just four percentage points higher than the calibration period (93%), the fraction of dry periods can be considered comparable to each other. However, the fraction of rainy periods for the calibration period (7%) is more than twice as high as for the validation period (3%). This implies that the calibration dataset is at least different with respect to the validation dataset concerning the percentage of rainy periods, which may have resulted in a lower MCC value for validation..."

**Section 3.2.1: In my opinion it should be pointed out here that the absolute number of false-positives is higher than the number of true positives. This is important for the interpretation of CML rainfall estimates because it means that more than 50% of the data points where CMLs estimate rainfall can be considered artifacts. As can be see in Polz et al. 2020 (https://doi.org/10.5194/amt-13-3835-2020) in Fig 9 this is not uncommon. The impact of the false-positives on the resulting rainfall amount is, however, smaller than the count of the false-positives suggest, as can be seen in in Polz et al. 2020 Fig 9d and 9f. In your case the impact of the false-positives on the rainfall amount might be different, though. Given the impact of false-positives on PBIAS in your analysis, the false-positive rain rates might play a larger role here. This should be discussed in more detail, maybe also in the conclusion section because the frequent occurrence and impact of false-positives seems to be a peculiar characteristic of CML rainfall estimates that all potential users or producers of CML QPE should be aware of.**

Reply:

We wrote:

L357-360: "...Approximately 50% of the rainy events are classified as dry, both for the calibrated and default parameter sets. Similar results were reached by Polz et al., (2020), however, the impact of false rain detection on the resulting rainfall amounts was found to be smaller than the relatively poor wet period classification suggested...".

L374-377: "...Due to overestimation observed by the PBIAS values, we can conclude that the significant number of false-positives (i.e., erroneous rainfall detection), plays an important role here. Polz et al. (2020) observed a different behavior, in the sense that even having a large number of false-positives was not translated into such an overestimation of the rainfall amounts...".

Moreover, Table 5 gives total interpretation of the false-positives impact on our analyses.

**Fig 5: Why did the results for the default parameters change compared to the same plot, Fig 4, in the initial submission? E.g. KGE is now 0.37 for the default parameters. It was 0.45 in the initial submission for the default parameters.**

Reply:

We redid the evaluation by using data.table package (https://github.com/Rdatatable/data.table) syntax, instead of pure R. Perhaps, this was the reason of difference in KGE values. The time aggregation performed by data.table works with a syntax near to SQL and proper for "big data" databases.

---

## Author Response (AR3)

**Author's replies to the referee's comments to manuscript AMT-2021-34**

**Please see the file with track changes to check the edited lines.**

The original referee's comments are written in bold and the author's replies are written in regular font.

**Anonymous Referee #2, revision 3**

**The authors have responded to all my comments and adjusted the manuscript accordingly. There are some minor details left that should be corrected, mainly regarding how the wet-dry classification results are discussed. See my comments below.**

We gratefully thank the Referee for the constructive comments and recommendations. We carefully checked the quality of the added text.

**L22: write "…by a delay…"**

Reply:

L22: "…This decrease was caused by a delay in data delivery…"

**L244: Since the authors acknowledge that they have no good explanation for the importance of WD_p2, they should add this information to the text. I suggest to write that there are some uncertainties (which I do not know of, but the authors should) regarding how the importance of the parameters are derived and that there is no clear explanation for the importance of WD_p2.**

Reply:

We added a sentence in 2.3 section in order to clarify how the importance of parameters are derived. Likely, using the right terminology let the things clearer.

L169-170 "…Thus, local sensitivities (i.e. a partial effects) get integrated o a global sensitivity measure…"

Moreover,

L241-244 "…It is important to highlight that the max( $P_{min}$ ) is only computed if at least a minimum number of hours (defined by $WD_{p1}$ ) of data are available; otherwise it is nor computed and no rainfall intensities are retrieved. Certainly, this results in uncertainties about the computation of sensitivity when rainfall is not retrieved. Thus, a clearer explanation for the highest importance of $WD_{p2}$ is not pointed.

**Fig. 2: The figure caption should explain what the number in the upper-right diagonal are. From the plot in the last revision it is clear to me that the first pair defines the range of the colours and I guess that the number behind is the correlation of the points within that range. But this is not clear to me, and might be even less clear to a reader that has not seen the previous version of this plot.**

Reply:

Thank you so much for this observation, we corrected the figure caption accordingly.

**L335: Better write "However, taking these parameters into account in the optimization..."**

Reply:
L325: "...However, taking these parameters into account in the optimization..."

**L346: "...for classified rainy events rightly..." sounds strange. If I understand correctly you mean the true positive rate. Please reformulate.**

Reply:

L340:"...However, the Sensitivity metric shows that the calibrated parameters are worst for classified the true positive rate..."

**L348 and following: In my opinion you do not have to use such negative formulations ("did not generalise at all"). It would suffice to point out that the optimization might not have generalized well enough. As you write in the text below the different fractions of wet events in the calibration and validation data set might also have contributed to this decrease of the MCC.**

Reply:

L340-342:"... This occurred because the optimization might not have generalized well enough the wet-dry classification process..."

**L357-360: Polz et al 2020 have not "reached" a result which is similar to, as you write, "approximately 50% of the rainy events are classified as dry". Their Fig 9 shows a much lower number of false-negatives compared to the true positives. Also Fig A1 clearly shows that there is approx. a 0.25 to 0.75 spilt between true-positives and false-negatives. Furthermore, I find these two new sentence a bit confusing because the first one seems to talk about false-negatives and the sec-**

**ond one about false-positives. Maybe this is only because the wording is not precise. I suggest to reformulate this new part slightly.**

Reply:

L354-355: "...Using a convolutional neural networks for classifying wet-dry periods, Polz et al. (2020) found a proportion of approximately 25% for false positives ..."

**L374-377: The logic of this new sentence is not correct. One cannot directly conclude that an overestimation (i.e. a high PBIAS) is caused by false-positives. Even if you have no false-positives and no false-negatives there could be bias, e.g. due to wrong WAA compensation or other effect. In my first two reviews I concluded that the false-positives have a strong effect on PBIAS because PBIAS changes dramatically if you do not include the false-positives in the analysis, as can be seen in Table 5. for the case "Reference > 0" which removes all cases where CMLs estimate rainfall but the reference is dry. Hence, I suggest to update this sentence and state that the results from Table 5 (as described above) indicate a strong effect of the false-positives on PBIAS. The fact that Polz et al 2020 do not see such a strong effect might be due to a lower number of false-positives in their analysis or due to a different distribution of false-positives. In their Fig 9 it is clear that the occurrence of false-positives has a peak at fairly small rain rates (0.1 mm/h) whereas the true-positives have the peak at 1 mm/h.**

Reply:

L385-387: "...Indeed, the false positives presented a strong effect on PBIAS, when they were removed (i.e., "Reference > 0") PBIAS changed significantly. As for Polz et al. (2020) a different behavior was observed, maybe due to a lower number of false positives or due to a different distribution of false positives..."